# Constant-Factor Approximation Algorithms for Socially Fair $k$-Clustering

## Abstract

We study approximation algorithms for the socially fair $(\ell_p, k)$-clustering problem with $m$ groups which include the socially fair $k$-median ($p = 1$) and $k$-means ($p = 2$). We present (1) a polynomial-time $(5 + 2\sqrt{6})^p$-approximation with at most $k + m$ centers (2) a $(5 + 2\sqrt{6} + \epsilon)^p$-approximation with $k$ centers in time $(nk)^{2^{O(p)}m^2/\epsilon}$, and (3) a $(15 + 6\sqrt{6})^p$ approximation with $k$ centers in time $k^m \cdot \text{poly}(n)$. The former is obtained by a refinement of the iterative rounding method via a sequence of linear programs. The latter two are obtained by converting a solution with up to $k + m$ centers to one with $k$ centers by sparsification methods for (2) and via an exhaustive search for (3). We also compare the performance of our algorithms with existing approximation algorithms on benchmark datasets, and find that our algorithms outperform existing methods.

## 1 Introduction

Automated decision making using machine learning algorithms is being widely adopted in modern society. Examples of real-world decision being made by ML algorithms are innumerable and include applications with considerable societal effects such as automated content moderation Gorwa et al. (2020) and recidivism prediction Angwin et al. (2016). This necessitates designing (new) machine learning algorithms that incorporate societal considerations, especially *fairness* Dwork et al. (2012); Kearns and Roth (2019).

The facility location problem is a well-studied problem in combinatorial optimization. Famous instances include the $k$-means, $k$-median and $k$-center problems, where the input is a finite metric and the goal is to find $k$ points ("centers" or "facilities") such that a function of distances of each given point to its nearest center is minimized. For $k$-means, the objective is the average squared distance to the nearest center; for $k$-median, it is the average distance; and for $k$-center, it is the maximum distance. These are all captured by the $(\ell_p, k)$-*clustering* problem, defined as follows: given a set of clients $\mathcal{A}$ of size $n$, a set of candidate facility locations $\mathcal{F}$, and a metric $d$, find a subset $F \subset \mathcal{F}$ of size $k$ that minimizes $\sum_{i \in \mathcal{A}} d(i, F)^p$, where $d(i, F) = \min_{j \in F} d(i, j)$. This is NP-hard for all $p$, and also hard to approximate Drineas et al. (2004); Guha and Khuller (1999). A $2^{O(p)}$-approximation algorithm was given by Charikar et al. (2002)[1]. The current best approximation factors for $k$-median and $k$-means on general metrics are $(2.675 + \epsilon)$-approximation Byrka et al. (2014) and $(9 + \epsilon)$-approximation Kanungo et al. (2004); Ahmadian et al. (2019), respectively.

Here we consider *socially fair* extensions of the $(\ell_p, k)$-clustering problem in which $m$ different (not necessarily disjoint) subgroups, $\mathcal{A} = A_1 \cup \cdots \cup A_m$, among the data are given, and the goal is to minimize the *maximum* cost over the groups, so that a common solution is not too expensive for any one of them. Each group can be a subset of the data or simply any nonnegative weighting. The goal is to minimize the maximum weighted cost among the groups, i.e.,

$$\min_{F \subset \mathcal{F}: |F| = k} \max_{s \in [m]} \sum_{i \in A_s} w_s(i) d(i, F)^p. \tag{1}$$

A weighting of $w_s(i) = 1/|A_s|$, for $i \in A_s$, corresponds to the average of groups. The groups usually arise from sensitive attributes such as race and gender (that are protected against discrimination

---

[1]In some other works, the $p$'th root of the objective is considered and therefore the approximation factors look different in such works.

under the Civil Rights Act of 1968 Hutchinson and Mitchell (2019); Benthall and Haynes (2019)). The cases of $p = 1$ and $p = 2$ are the socially fair $k$-median and $k$-means, respectively, introduced by Ghadiri et al. (2021); Abbasi et al. (2021). As discussed in Ghadiri et al. (2021), the objective of the socially fair $k$-means promotes a more equitable average clustering cost among different groups.

The objective function of socially fair $k$-median was first studied by Anthony et al. (2010) who gave an $O(\log m + \log n)$-approximation algorithm. Moreover, the existing approximation algorithms for the vanilla $k$-means and $k$-median can be used to find $O(m)$-approximate solutions for the socially fair versions Ghadiri et al. (2021); Abbasi et al. (2021). The proof technique directly yields a $m \cdot 2^{O(p)}$-approximation for the socially fair $(\ell_p, k)$-clustering. The natural linear programming (LP) relaxation of the socially fair $k$-median problem has an integrality gap of $\Omega(m)$ Abbasi et al. (2021).

More recently, Makarychev and Vakilian (2021) strengthened the LP relaxation of the socially fair $(\ell_p, k)$-clustering by a sparsification technique. The stronger LP has an integrality gap of $\Omega(\log m / \log \log m)$ and their rounding algorithm (similar to Charikar et al. (2002)) finds a $(2^{O(p)} \log m / \log \log m)$-approximation algorithm for the socially fair $(\ell_p, k)$-clustering. For the socially fair $k$-median, this is asymptotically the best possible in polynomial time under the assumption NP $\not\subseteq \bigcap_{\delta > 0} \text{DTIME}(2^{n^\delta})$ Bhattacharya et al. (2014). Due to this hardness result, it is natural to consider a *bicriteria* approximation, which allows for more centers whose total cost is close to the optimal cost for $k$ centers. For the socially fair $k$-median and $0 < \epsilon < 1$, Abbasi et al. (2021) presents an algorithm that gives at most $k/(1-\epsilon)$ centers with objective value at most $2^{O(p)}/\epsilon$ times the optimum for $k$ centers. Our first result is an improved bicriteria approximation algorithm for the socially fair $\ell_p$ clustering problem with only $m$ additional centers ($m$ is usually a small constant).

**Theorem 1.1.** *There is a polynomial-time bicriteria approximation algorithm for the socially fair $(\ell_p, k)$-clustering problem with $m$ groups that finds a solution with at most $k + m$ centers of cost at most $(5 + 2\sqrt{6})^p \approx 9.9^p$ times the optimal cost for a solution with $k$ centers.*

Goyal and Jaiswal Goyal and Jaiswal (2021) show that a solution to the socially fair $(\ell_p, k)$-clustering problem with $k' > k$ centers and cost $C$ can be converted to a solution with $k$ centers and cost at most $3^{p-1}(C + 2\text{opt})$ by simply taking the $k$-subset of the $k'$ centers of lowest cost. A proof is in the appendix for completeness. We improve this factor using a sparsification technique.

**Theorem 1.2.** *For any $\epsilon > 0$, there is a $(5 + 2\sqrt{6} + \epsilon)^p$-approximation algorithm for the socially fair $(\ell_p, k)$-clustering problem that runs in time $(nk)^{2^{O(p)} m^2 / \epsilon}$; there is a $(15 + 6\sqrt{6})^p$-approximation algorithm that runs in time $k^m \cdot poly(n)$.*

This raises the question of whether a faster-constant-factor approximation is possible. Goyal and Jaiswal (2021) show under the Gap-Exponential Time Hypothesis[2], it is hard to approximate socially fair $k$-median and $k$-means within factors of $1 + 2/e - \epsilon$ and $1 + 8/e - \epsilon$, respectively, in time $g(k) \cdot n^{f(m) \cdot o(k)}$, for $f, g : \mathbb{R}^+ \to \mathbb{R}^+$; socially fair $(\ell_p, k)$-clustering is hard to approximate within a factor of $3^p - \epsilon$ in time $g(k) \cdot n^{o(k)}$. They also give a $(3 + \epsilon)^p$-approximation in time $(k/\epsilon)^{O(k)} \text{poly}(n/\epsilon)$. This leaves open the possibility of a constant-factor approximation in time $f(m)\text{poly}(n, k)$.

For the case of $p \to \infty$, the problem reduces to fair $k$-center problem if we take $p^{\text{th}}$ root of the objective. The problem is much better understood and widely studied along with many generalization Jia et al. (2021); Anegg et al. (2021); Makarychev and Vakilian (2021). Makarychev and Vakilian (2021) result implies an $O(1)$-approximation in this case.

We compare the performance of our bicriteria algorithm against Abbasi et al. (2021) and our algorithm with exactly $k$ centers against Makarychev and Vakilian (2021) on three different benchmark datasets. Our experiments show that our algorithms consistently outperform these in practice (Section 5) and often select fewer centers than the algorithm of Abbasi et al. (2021) (Section E.3).

## 1.1 Approach and Techniques

Our starting point is a LP relaxation of the problem. The integrality gap of the natural LP relaxation is $m$ Abbasi et al. (2021). For our bicriteria result, we use an iterative rounding procedure, inspired

---

[2]Informally Gap-ETH states that there is no $2^{o(n)}$-time algorithm to distinguish between a satisfiable formula and a formula that is not even $(1 - \epsilon)$ satisfiable.

by Krishnaswamy et al. (2018). In each iteration, we solve an LP whose constraints change from one iteration to the next. We show that the feasible region of the final LP is the intersection of a matroid polytope and $m$ affine spaces. This implies that the size of the support of an optimal extreme solution is at most $k + m$ — see Lemma 1.3. Rounding up all of these fractional variables results in a solution with $k + m$ centers.

There are two approaches to convert a solution with up to $k + m$ centers to a solution with $k$ centers. The first is to take the best $k$-subset of the $k + m$ centers which results in a $(15 + 6\sqrt{6})^p$-approximation for an additional cost of $O(k^m n(k + m))$ in the running time. This follows from the work of Goyal and Jaiswal (2021). For completeness, we include it as Lemma A.1 in the Appendix.

The second approach is to "sparsify" the given instance of the problem. We show if the instance is "sparse," then the integrality gap of the LP is small. A similar idea was used by Li and Svensson (2016) for the classic $k$-median problem. We extend this sparsification technique to socially fair clustering. We define an $\alpha$-sparse instance for the socially fair $k$-median problem as an instance in which for an optimum set of facilities $O$, any group $s \in [m]$ and any facility $i$ not in the optimum solution, the number of clients of group $s$ in a ball of radius $d(i, O)/3$ centered at $i$ is less than $\frac{3\alpha|A_s|}{2d(i,O)}$. For such an instance, given a set of facilities, replacing facility $i$ with the closest facility to $i$ in $O$ can only increase the total cost of the clients served by this facility by a constant factor plus $2\alpha$. We show that if an instance is $O(\frac{\text{opt}}{m})$-sparse, then the integrality gap of the LP is constant.

For an $O(\texttt{opt}/m)$-sparse instance of the socially fair $k$-median problem, a solution with $k + m$ centers can be converted to a solution with $k$ centers in time $n^{O(m^2)}$ while increasing the objective value only by a constant factor. Our conversion algorithm is based on the fact that there are at most $O(m^2)$ facilities that are far from the facilities in the optimal solution. We enumerate candidates for these facilities and then solve an optimization problem for the facilities that are close to the facilities in the optimal solution. This optimization step is again over the intersection of the polytope of a matroid with $m$ half-spaces. In summary, our algorithm consists of three main steps.

1. We produce $n^{O(m^2)}$ instances of the problem such that at least one is $O(\frac{\text{opt}}{m})$-sparse and its optimal objective value is equal to that of the original instance (Section 4, Lemma 3.2).
2. For each of the instances produced in the previous step, we find a *pseudo-solution* with at most $k + m$ centers by an iterative rounding procedure (Section 2, Lemma 2.1).
3. We convert each pseudo-solution with $k + m$ centers to a solution with $k$ centers (Section 4, Lemma 4.2) and return the solution with the minimum cost.

## 1.2 PRELIMINARIES

We use terms *centers* and *facilities* interchangeably. For a set $S$ and item $i$, we denote $S \cup \{i\}$ by $S + i$. For sets $S_1, \ldots, S_k$, we denote their Cartesian product by $\bigotimes_{j \in [k]} S_j$, i.e., $(s_1, \ldots, s_k) \in \bigotimes_{j \in [k]} S_j$ if and only if $s_1 \in S_1, \ldots, s_k \in S_k$. For an instance $\mathcal{I}$ of the problem, we denote an optimal solution of $\mathcal{I}$ and its objective value by $\text{OPT}_{\mathcal{I}}$ and $\text{opt}_{\mathcal{I}}$, respectively. A pair $\mathcal{M} = (E, \mathcal{I})$, where $\mathcal{I}$ is a non-empty family of subsets of $E$, is a matroid if: 1) for any $S \subseteq T \subseteq E$, if $T \in \mathcal{I}$ then $S \in \mathcal{I}$ (hereditary property); and 2) for any $S, T \in \mathcal{I}$, if $|S| < |T|$, then there exists $i \in T \setminus S$ such that $S + i \in \mathcal{I}$ (exchange property) see Schrijver (2003). We call $\mathcal{I}$ the set of independent sets of the matroid $\mathcal{M}$. The basis of $\mathcal{M}$ are all the independent sets of $\mathcal{M}$ of maximal size. The size of all of the basis of a matroid is equal and is called the *rank* of the matroid. We use the following lemma in the analysis of both our bicriteria algorithm and the algorithm with exactly $k$ centers.

**Lemma 1.3.** *Grandoni et al. (2014) Let $\mathcal{M} = (E, \mathcal{I})$ be a matroid with rank $k$ and $P(\mathcal{M})$ denote the convex hull of all basis of $\mathcal{M}$. Let $Q$ be the intersection of $P(M)$ with $m$ additional affine constraints. Then any extreme point of $Q$ has a support of size at most $k + m$.*

**Related work.** Unsupervised learning under fairness constraints has received significant attention over the past decade. Social fairness (i.e., equitable cost for demographic groups) has been considered for problems such as PCA Samadi et al. (2018); Tantipongpipat et al. (2019). Other notions of fairness are also considered for clustering. The most notables are balance in clusters (i.e., equitable representation of groups) Chierichetti et al. (2017); Abraham et al. (2020); Bera et al. (2019); Ahmadian et al. (2019), balance in representation (i.e., equitable representation of groups in selected centers) Hajiaghayi et al. (2010); Krishnaswamy et al. (2011); Kleindessner et al. (2019), and individual fairness Kleindessner et al. (2020); Vakilian and Yalciner (2022); Chakrabarti et al. (2022).

However as shown by Gupta et al. (2020), different notions of fairness are incompatible in the sense that they cannot be satisfied simultaneously. For example, see the discussion and experimental result regarding incompatibility of social fairness and equitable representation in Ghadiri et al. (2021). In addition, several other notions of fairness for clustering has been considered Chen et al. (2019); Jung et al. (2020); Mahabadi and Vakilian (2020); Micha and Shah (2020); Brubach et al. (2020).

## 2 BICRITERIA APPROXIMATION

In this section, we prove Theorem 1.1. Our method relies on solving a series of linear programs and utilizing the iterative rounding framework for the $k$-median problem as developed in Krishnaswamy et al. (2018); Gupta et al. (2021). We aim for a cleaner exposition over smaller constants below. We use the following standard linear programming (LP) relaxation (**LP1**).

$$
\begin{array}{ll}
\min \ z & \textbf{(LP1)} \\
\text{s.t.} \ z \geq \sum_{i \in A_s, j \in \mathcal{F}} w_s(i) d(i,j)^p x_{ij}, \\
\qquad\qquad\qquad\qquad \forall \, 1 \leq s \leq m, \\
x_{ij} \leq y_j \quad, \forall \, i \in \mathcal{A}, j \in \mathcal{F}, \\
\sum_{j \in \mathcal{F}} y_j = k, \\
\sum_{j \in \mathcal{F}} x_{ij} = 1 \quad, \forall \, i \in \mathcal{A}, \\
x, y \geq 0.
\end{array}
\qquad
\begin{array}{ll}
\min \ z & \textbf{(LP2)} \\
\text{s.t.} \ z \geq \sum_{j \in A_s} \sum_{i \in F_j} w_s(i) d(i,j)^p y_i, \\
\qquad\qquad\qquad\qquad \forall \, 1 \leq s \leq m, \ (2) \\
\\
\sum_{j \in \overline{\mathcal{F}}} y_j = k, \hfill (3) \\
\sum_{j \in F_i} y_j = 1 \quad, \forall i \in \mathcal{A}, \hfill (4) \\
y \geq 0.
\end{array}
$$

Theorem 1.1 follows as a corollary to Lemma 2.1 as we can pick all the fractional centers integrally. Observe that, once the centers are fixed, the optimal allocation of clients to facilities is straightforward: every client connects to the nearest opened facility.

**Lemma 2.1.** *Let $0 < \lambda \leq 1$. There is a polynomial time algorithm that given a feasible solution $(\tilde{x}, \tilde{y}, \tilde{z})$ to the linear program **LP1** returns a feasible solution $(x', y', z')$ where $z' \leq ((1+2(1+\lambda)/\lambda)(1+\lambda))^p \tilde{z}$ and the size of the support of $y'$ is at most $k+m$. The running time is polynomial in $n$ and the logarithm of the distance of the farthest points divided by $\lambda$.*

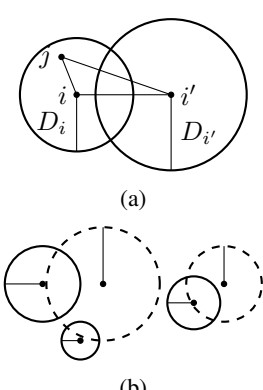

(a)

(b)

Figure 1: (a) Distance of $i'$ from the facilities of its representative $i$. (b) Solid and dashed circles are the balls corresponding to representative $(\mathcal{U}^\star)$ and non-representative clients $(\mathcal{U}^f)$, respectively.

*Proof.* We describe the iterative rounding argument to round the solution $(\tilde{x}, \tilde{y}, \tilde{z})$. As a first step, we work with an equivalent linear program **LP2**, where we have removed the assignment variables $x$. This can be achieved by splitting each facility $j$ to the number of unique nonzero $\tilde{x}_{ij}$'s and setting the corresponding variable for these facilities accordingly, e.g., if the unique weights are $\tilde{x}_{1j} < \tilde{x}_{2j} < \tilde{x}_{3j}$, then the corresponding weights for the facilities are $\tilde{x}_{1j}$, $\tilde{x}_{2j} - \tilde{x}_{1j}$, and $\tilde{x}_{3j} - \tilde{x}_{2j}$ and the weights of the connections between these new facilities and clients are determined accordingly as either zero or the weight of the facility.

Let $\overline{\mathcal{F}}$ be the set of all (splitted) copies of facilities. Then we can assume $\tilde{x}_{ij} \in \{0, \tilde{y}_j\}$ for each $i, j$ (where $j \in \overline{\mathcal{F}}$). We set $F_i = \{j \in \overline{\mathcal{F}} : \tilde{x}_{ij} > 0\}$. Note that $F_i$ could contain multiple copies of original facilities. Observe that **LP2** has a feasible solution $(\tilde{y}, \tilde{z})$ for this choice of $F_i$ for each $i \in \mathcal{A}$. Moreover, any feasible solution to **LP2** can be converted to a solution of **LP1** of same cost while ensuring that each client $i$ gets connected to the original copy of the facilities in $F_i$.

The iterative argument is based on the following. We group nearby clients and pick only one *representative* for each group such that if each client is served by the facility that serves its representative, the cost is at most $((1+2(1+\lambda)/\lambda)(1+\lambda))^p \tilde{z}$. Moreover, we ensure that candidate facilities $F_i$ for representative clients are disjoint. In this case, one observes that the constraints Eq. 3-Eq. 4 in **LP2** define the convex hull of a partition matroid and must be integral. Indeed, this already gives an integral solution to the basic $k$-median problem. But, in the socially fair clustering problem, there are $m$ additional constraints, one for each of the $m$ groups. Nevertheless, by Lemma 1.3, any extreme point solution to the matroid polytope intersected with at most $m$ linear constraints has a support of size at most $k + m$ (see also Lau et al. (2011) Chap. 11).

---

**Algorithm 1:** Iterative Rounding

1 **Input:** $\mathcal{A} = A_1 \cup \cdots \cup A_m, \mathcal{F}, k, d, \lambda$
2 **Output:** A set of centers of size at most $k + m$.
3 Solve **LP1** to get optimal solution $(x^\star, y^\star, z^\star)$ and reate set $\overline{\mathcal{F}}$ by splitting facilities.
4 Set $F_i = \{j \in \overline{\mathcal{F}} : x^\star_{ij} > 0\}, D_i = \max\{d_{ij} : x^\star_{ij} > 0\}$ for each $i \in \mathcal{U}$.
5 Sort clients in increasing order of $D_i$ and greedily include clients in $\mathcal{U}^\star$ while maintaining that $\{F_i : i \in \mathcal{U}^\star\}$ remain disjoint.
6 Set $\mathcal{U}^f = \mathcal{A} \setminus \mathcal{U}^\star$.
7 **while** *there is some tight constraint from Eq. 8* **do**
8     **if** *there exists $i \in \mathcal{U}^f$ such that $y(B_i) = 1$ (i.e., Eq. 8 is tight for $i$)* **then**
9

$$F_i \leftarrow B_i, D_i \leftarrow \frac{D_i}{1+\lambda}, B_i \leftarrow \{j \in F_i : d(i,j) \leq \frac{D_i}{1+\lambda}\}, \text{ and } \text{Update-}\mathcal{U}^\star(i).$$

10     Find an extreme point solution $y$ to the linear program LP($\mathcal{U}^\star, \mathcal{U}^f, D$) .
11 **return** the support of $y$ in the solution of LP($\mathcal{U}^\star, \mathcal{U}^f, D$) .
12

---

1 **Procedure** *Update-$\mathcal{U}^\star(i)$*
2     **if** *for every $i' \in \mathcal{U}^\star$ that $F_i \cap F_{i'} \neq \emptyset$, $D_{i'} > D_i$* **then**
3        Remove all $i'$ that represent $i$ from $\mathcal{U}^\star$ and add them to $\mathcal{U}^f$.
4        $\mathcal{U}^\star \leftarrow \mathcal{U}^\star \cup \{i\}$.

---

We now formalize the argument and specify how one iteratively groups the clients. We iteratively remove/change constraints in **LP2** as we do the grouping while ensuring that linear program's cost does not increase. We initialize $D_i = \max\{d(i,j) : \tilde{x}_{ij} > 0\} = \max\{d(i,j) : j \in F_i\}$ for each client $i$. We maintain a set of representative clients $\mathcal{U}^\star$. We say a client $i \in \mathcal{U}^\star$ represents a client $i'$ if they share a facility, i.e., $F_i \cap F_{i'} \neq \emptyset$, and $D_i \leq D_{i'}$. The representative clients do not share any facility with each other. The non-representative clients are put in the set $\mathcal{U}^f$. We initialize $\mathcal{U}^\star$ as follows. Sort all clients in increasing order of $D_i$. Greedily add clients to $\mathcal{U}^\star$ while maintaining $F_i \cap F_{i'} = \emptyset$ for all $i \neq i' \in \mathcal{U}^\star$. Observe that $\mathcal{U}^\star$ is maximal with above property, i.e., for every $i' \notin \mathcal{U}^\star$, there is $i \in \mathcal{U}^\star$ such that $F_i \cap F_{i'} \neq \emptyset$ and $D_i \leq D_{i'}$. We will maintain this invariant in the algorithm. For clients $i \in \mathcal{U}^f$, we set $B_i$ to be the facilities in $F_i$ that are within a distance of $\frac{D_i}{1+\lambda}$.

In each iteration we solve the following linear program and update $\mathcal{U}^\star, \mathcal{U}^f, D_i$'s, $B_i$'s, and $F_i$'s.

$$\min z \qquad\qquad\qquad (\text{LP}(\mathcal{U}^\star, \mathcal{U}^f, D))$$

$$\text{s.t. } z \geq \sum_{i \in A_s \cap \mathcal{U}^\star} w_i(s) \sum_{j \in F_i} d(i,j)^p y_j$$

$$+ \sum_{i \in A_s \cap \mathcal{U}^f} w_i(s) \left( \sum_{j \in B_i} d(i,j)^p y_j + (1 - y(B_i))D_i^p \right), \forall 1 \leq s \leq m, \qquad (5)$$

$$\sum_{j \in \overline{\mathcal{F}}} y_j = k, \qquad\qquad\qquad (6)$$

$$\sum_{j \in F_i} y_j = 1 \quad , \forall i \in \mathcal{U}^\star, \qquad\qquad (7)$$

$$\sum_{j \in B_i} y_j \leq 1 \quad , \forall i \in \mathcal{U}^f, \qquad\qquad (8)$$

$$y \geq 0. \qquad\qquad\qquad (9)$$

For clients in $i \in \mathcal{U}^f$, we only insist that we pick *at most* one facility from $B_i$ (see Eq. 8). The objective is modified to pay $D_i$ for any fractional shortfall of facilities in this smaller ball (see Eq. 5). Observe that if this additional constraint Eq. 8 becomes tight for some $j \in \mathcal{U}^f$, we can decrease $D_i$ by a factor of $(1 + \lambda)$ for this client and then update $\mathcal{U}^\star$ accordingly to see if $i$ can be included in it. Also, we round each $d(i,j)$ to the nearest power of $(1 + \lambda)$. This only changes the objective by a factor of $(1 + \lambda)^p$ and we abuse notation to assume that $d$ satisfies this constraint (it might no longer be a metric but in the final assignment, we will work with its metric completion). The iterative algorithm runs as described in Algorithm 1. It is possible that a client moves between $\mathcal{U}^f$ and $\mathcal{U}^\star$ but any time that a point is processed in $\mathcal{U}^f$ (Step 3(a) above), $D_i$ is divided by $(1 + \lambda)$. Thus the algorithm takes $O(n \log \frac{(\text{diam})}{\lambda})$ iterations, where diam is the distance between the two farthest points. Finally the result is implied by the following claims (proved in the appendix).

**Claim 2.2.** *The cost of the LP is non-increasing over iterations. Moreover, when the algorithm ends, there are at most $k + m$ facilities in the support.*

---
**Algorithm 2:** Sparsify
---
1 **Input:** $\mathcal{A} = A_1 \cup \cdots \cup A_m, \mathcal{F}, k, d, t \in \mathbb{N}$
2 **Output:** A set of fair $k$-median instances.
3 **for** $t' = 1, \ldots, m^2 t$ *and* $t'$ ***facility pairs*** $(j_1, j_1'), \ldots, (j_{t'}, j_{t'}')$ **do**
4  $\quad$ Output $\mathcal{I}' = (\mathcal{F}', \mathcal{A}, k, d)$, where $\mathcal{F}' = \mathcal{F} \setminus \bigcup_{r=1}^{t'} \texttt{FBALL}(j_r, d(j_r, j_r'))$.

---

**Claim 2.3.** *For any client* $i' \in \mathcal{U}^f$*, there is always one total facility at a distance of at most* $(1 + 2(1+\lambda)/\lambda) D_{i'}$*, i.e.,* $\sum_{j:d(i',j) \leq (1+2(1+\lambda)/\lambda)D_{i'}} y_j \geq 1$.

**Claim 2.4.** *Let* $\hat{y}$ *be an integral solution to linear program* $LP(\mathcal{U}^\star, \mathcal{U}^f, D)$ *after the last iteration. Then, there is a solution* $(\hat{x}, \hat{y})$ *to the linear program* **LP1** *such that objective is at most* $((1+2(1+\lambda)/\lambda)(1+\lambda))^p$ *times the objective of the linear program* $LP(\mathcal{U}^\star, \mathcal{U}^f, D)$ *.* $\quad\square$

Now we prove Theorem 1.1 by substituting the best $\lambda$ in Lemma 2.1.

*Proof of Theorem 1.1.* By Lemma 2.1, the output vector of Algorithm 1 corresponding to the centers has a support of size at most $k + m$. Rounding up all the fractional centers, we get a solution with at most $k + m$ centers and a cost of at most $((1+2(1+\lambda)/\lambda)(1+\lambda))^p$ of the optimal. We optimize over $\lambda$ by taking the gradient of $(1+2(1+\lambda)/\lambda)(1+\lambda)$ and setting it to zero. This gives the optimum value of $\lambda = \sqrt{2/3}$. Substituting this, gives a total approximation factor of $(5 + 2\sqrt{6})^p$. $\quad\square$

## 3 Approximation Algorithms for Fair $k$-Clustering

We first show how to generate a set of instances such that at least one of them is sparse and has the same optimal objective value as the original instance. Then we present our algorithm to find a solution with $k$ facilities from a pseudo-solution with $k + m$ facilities for a sparse instance, inspired by Li and Svensson (2016). We need to address new difficulties: the sparsity with respect to all groups $s \in [m]$; and as our pseudo-solution has $m$ additional centers (instead of $O(1)$ additional centers), we need a sparser instance compared to Li and Svensson (2016). One new technique is solving the optimization problem given in Step 11 of Algorithm 3 (Lemma 4.1). This is trivial for the vanilla $k$-median but in the fair setting, we use certain properties of the extreme points of intersection of a matroid polytope with half-spaces, and combine this with a careful enumeration.

For an instance $\mathcal{I}$, we denote the cost of a set of facilities $F$ by $\texttt{cost}_\mathcal{I}(F)$. For a point $q$ and $r > 0$, we denote the facilities in the ball of radius $r$ at $q$ by $\texttt{FBall}_\mathcal{I}(q, r)$. This does not contain facilities at distance exactly $r$ from $q$. For a group $s \in [m]$, the set of clients of $s$ in the ball of radius $r$ at $q$ is $\texttt{CBall}_{\mathcal{I},s}(q, r)$. Note that because we consider the clients as weights on points, $\texttt{CBall}_{\mathcal{I},s}(q, r)$ is actually a set of (point, weight) pairs. We let $|\texttt{CBall}_{\mathcal{I},s}(q, r)| = \sum_{i \in \texttt{CBall}_{\mathcal{I},s}(q,r)} w_s(i)$.

**Definition 3.1.** *[Sparse Instance] For* $\alpha > 0$*, an instance of the fair* $\ell_p$ *clustering problem* $\mathcal{I} = (k, \mathcal{F}, \mathcal{A}, d)$ *is* $\alpha$-*sparse if for each facility* $j \in \mathcal{F}$ *and group* $s \in [m]$,

$$\left(\tfrac{2}{3} d(j, OPT_\mathcal{I})\right)^p \cdot |\texttt{CBall}_{\mathcal{I},s}(j, \tfrac{1}{3} d(j, OPT_\mathcal{I}))| \leq \alpha.$$

*We say that a facility* $j$ *is* $\alpha$-*dense if it violates the above for some group* $s \in [m]$.

To motivate the definition, let $\mathcal{I}$ be an $\alpha$-sparse instance, $OPT_\mathcal{I}$ be an optimal solution of $\mathcal{I}$, $j$ be a facility not in $OPT_\mathcal{I}$ and $j^*$ be the closest facility in $OPT_\mathcal{I}$ to $j$. Let $F$ be a solution that contains $j$ and $\eta_{j,s}$ be the total cost of the clients of group $s \in [m]$ that are connected to $j$ in solution $F$. Then,

(cost of group $s$ for solution $F \cup j \setminus j^*$) $\leq$ (cost of group $s$ for solution $F$) $+ 2^{O(p)} \cdot (\alpha + \eta_{j,s})$.

This property implies that if $\alpha \leq \texttt{opt}_\mathcal{I}/m$, then replacing $m$ different facilities can increase the cost by a factor of $2^{O(p)}$ plus $2^{O(p)} \cdot \texttt{opt}_\mathcal{I}$, and the integrality gap of the LP relaxation is $2^{O(p)}$. The next algorithm generates a set of instances such that at least one of them has objective value equal to $\texttt{opt}_\mathcal{I}$ and is $(\texttt{opt}_\mathcal{I}/mt)$-sparse for a fixed integer $t$.

**Lemma 3.2.** *Algorithm 2 runs in* $n^{O(m^2 t)}$ *time and produces instances of the socially fair* $\ell_p$ *clustering problem such that at least one of them satisfies the following: (1) The optimal value of the original instance* $\mathcal{I}$ *is equal to the optimal value of the produced instance* $\mathcal{I}'$*; (2)* $\mathcal{I}'$ *is* $\frac{\texttt{opt}_\mathcal{I}}{mt}$-*sparse.*

# 4 CONVERTING A SOLUTION WITH $k+m$ CENTERS TO ONE WITH $k$ CENTERS

We first analyze the special case when the set of facilities is partitioned to $k$ disjoint sets and we are constrained to pick exactly one from each set. This will be a subroutine in our algorithm.

**Lemma 4.1.** *Let $S_1, \ldots, S_k$ be disjoint sets such that $S_1 \cup \cdots \cup S_k = [n]$. For $g \in [m]$, $j \in [k]$, $v \in S_j$, let $\alpha_v^{(g,j)} \geq 0$. Then there is an $(nk)^{O(m^2/\epsilon)}$-time algorithm that finds a $(1+\epsilon)$-approximate solution to $\min_{v_i \in S_i : i \in [k]} \max_{g \in [m]} \sum_{j \in [k]} \alpha_{v_j}^{(g,j)}$.*

*Proof.* The LP relaxation of the above problem is

$$\min \; \theta \quad \text{such that } \sum_{j \in [k]} \sum_{v \in S_j} \alpha_v^{(g,j)} x_v^{(j)} \leq \theta, \; \forall g \in [m],$$
$$\sum_{v \in S_j} x_v^{(j)} = 1, \; \forall j \in [k],$$
$$x^{(j)} \geq 0, \; \forall j \in [k].$$

Note that this is equivalent to optimizing over a partition matroid with $m$ extra linear constraints. Therefore by Lemma 1.3, an extreme point solution has a support of size at most $k + m$. Now suppose $\theta^*$ is the optimal integral objective value, and $v_1^* \in S_1, \ldots, v_k^* \in S_k$ are the points that achieve this optimal objective. For each $g \in [m]$, at most $m/\epsilon$ many $\alpha_{v_j^*}^{(g,j)}$, $j \in [k]$, can be more than $\frac{\epsilon}{m}\theta^*$ because $\sum_{j \in [k]} \alpha_{v_j^*}^{(g,j)} \leq \theta^*$. Suppose, for each $g \in [m]$, we guess the set of indices $T_g = \{j \in [k] : \alpha_{v_j^*}^{(g,j)} \geq \frac{\epsilon}{m}\theta^*\}$. This takes $k^{O(m^2/\epsilon)}$ time by enumerating over set $[k]$. Let $T = T_1 \cup \cdots \cup T_m$. For $j \in T$, we also guess $v_j^*$ in the optimum solution by enumerating over $S_j$'s, $j \in T$. This increases the running time by a multiplicative factor of $n^{O(m^2/\epsilon)}$ since $v_j^* \in S_j$ and $|S_j| \leq n$. Based on these guesses, we can set the corresponding variables in the LP, i.e., for each $S_j$ such that $j \in T$, we add the following constraints for $v \in S_j$. $x_v^{(j)} = 1$ if $v = v_j^*$, and $x_v^{(j)} = 0$, otherwise. The number of LPs generated by this enumeration is $(nk)^{O(m^2/\epsilon)}$.

We solve all such LPs to get optimum extreme points. Let $(\bar{\theta}, \bar{x}^{(1)}, \ldots, \bar{x}^{(k)})$ be an optimum extreme point for the LP corresponding to the right guess (i.e., the guess in which we have identified all indices $j \in [k]$ along with their corresponding $v_j^*$ such that there exists $g \in [m]$ where $\alpha_{v_j^*}^{(g,j)} \geq \frac{\epsilon}{m}\theta^*$). Therefore $\bar{\theta} \leq \theta^*$. Let $R = \{j \in [k] : \bar{x}^{(j)} \notin \{0,1\}^{|S_j|}\}$. Since the feasible region of the LP corresponds to the intersection of a matroid polytope and $m$ half-spaces, by Lemma 1.3, the size of the support of an extreme solution is $k + m$. Moreover any cluster with $j \in [k]$ that have fractional centers in the extreme point solution, contributes at least 2 to the size of the support because of the equality constraint in the LP. Therefore $2|R| + (k - |R|) \leq k + m$ which implies $|R| \leq m$.

Now we guess the $v_j^*$ for all $j \in R$. By construction, $R \cap T = \emptyset$. Therefore for all $j \in R$ and $g \in [m]$, $\alpha_{v_j^*}^{(g,j)} < \frac{\epsilon}{m}\theta^*$. Therefore for all $g \in [m]$, $\sum_{j \in R} \alpha_{v_j^*}^{(g,j)} \leq m \cdot \frac{\epsilon}{m}\theta^* = \epsilon\theta^*$. Thus for the right guess of $v_j^*$, $j \in R$, we get an integral solution with a cost less than or equal to $\bar{\theta} + \epsilon\theta^* \leq (1 + \epsilon)\theta^*$. $\square$

Algorithm 3 is our main procedure to convert a solution with $k + m$ centers to a solution with $k$ centers. We need $\beta$ to be in the interval mentioned in Lemma 4.2. To achieve this we guess $\text{opt}_{\mathcal{I}}$ as different powers of two and try the corresponding $\beta$'s. The main idea behind the algorithm is that in a pseudo-solution of a sparse instance, there are only a few ($< m^2 t$) facilities that are far from facilities in the optimal solution. So the algorithm tries to guess those facilities and replace them by facilities in the optimal solution. For the rest of the facilities in the pseudo-solution (which are close to facilities in the optimal solution), the algorithm solves an optimization problem (Lemma 4.1) to find a set of facilities with a cost comparable to the optimal solution.

Finally combining the following lemma (proved in the appendix) with Lemma 3.2 (sparsification) and Theorem 1.1 (bicriteria algorithm) implies Theorem 1.2.

**Lemma 4.2.** *Let $\mathcal{I} = (k, \mathcal{F}, \mathcal{A}, d)$ be an $\frac{\text{opt}_{\mathcal{I}}}{mt}$-sparse instance of the $(\ell_p, k)$-clustering problem, $\mathcal{T}$ be a pseudo-solution with at most $k + m$ centers, $\epsilon' > 0$, $\delta \in (0, \min\{\frac{1}{8}, \frac{\log(1+\epsilon')}{12}\})$, $t \geq 4(1 + \frac{3}{\delta})^p$*

---

**Algorithm 3:** Obtaining a solution from a pseudo-solution

1    **Input:** Instance $\mathcal{I}$, $\beta$, a pseudo-solution $\mathcal{T}$, $\epsilon' > 0$, $\delta \in (0, \min\{\frac{1}{8}, \frac{\log(1+\epsilon')}{12}\})$, and integer $t \geq 4 \cdot (1 + \frac{3}{\delta})^p$.

2    **Output:** A solution with at most $k$ centers.

3    $\mathcal{T}' \leftarrow \mathcal{T}$

4    **while** $|\mathcal{T}'| > k$ *and there is* $j \in \mathcal{T}'$ *s.t.* $\mathtt{cost}_\mathcal{I}(\mathcal{T}' \setminus \{j\}) \leq \mathtt{cost}_\mathcal{I}(\mathcal{T}') + \beta$ **do** $\mathcal{T}' \leftarrow \mathcal{T}' \setminus \{j\}$;

5    **if** $|\mathcal{T}'| \leq k$ **then return** $\mathcal{T}'$;

6    **forall** $\mathcal{D} \subseteq \mathcal{T}'$ *and* $\mathcal{V} \subseteq \mathcal{F}$ *such that* $|\mathcal{D}| + |\mathcal{V}| = k$ *and* $|\mathcal{V}| < m^2 t$ **do**

7       For $j \in \mathcal{D}$, set $L_j = d(j, \mathcal{T}' \setminus \{j\})$

8       For $s \in [m]$, $j \in \mathcal{D}$, $f_j \in \mathtt{FBall}_\mathcal{I}(j, \delta L_j)$, set $\alpha_{f_j}^{(s,j)} = \displaystyle\sum_{i \in \mathtt{CBall}_{\mathcal{I},s}(j, L_j/3)} \min\{d(i, f_j)^p, d(i, \mathcal{V})^p\}$.

9       Let $(\widetilde{f}_j : j \in \mathcal{D}) \in \bigotimes_{j \in D} \mathtt{FBall}_\mathcal{I}(j, \delta L_j)$ be $(1 + \epsilon)$-approximate solution to (see Lemma 4.1)

$$\min_{(f_j : j \in \mathcal{D}) \in \bigotimes_{j \in \mathcal{D}} \mathtt{FBall}_\mathcal{I}(j, \delta L_j)} \max_{s \in [m]} \sum_{j' \in \mathcal{D}} \alpha_{f_{j'}}^{(s,j')}$$

      $\mathcal{S}_{\mathcal{D}, \mathcal{V}} \leftarrow \mathcal{V} \cup \{\widetilde{f}_j : j \in \mathcal{D}\}$

10   **return** $\mathcal{S} := \arg\min_{\mathcal{S}_{\mathcal{D}, \mathcal{V}}} \mathtt{cost}_\mathcal{I}(\mathcal{S}_{\mathcal{D}, \mathcal{V}})$

---

*be an integer, and* $\frac{2}{mt}\left(\mathtt{opt}_\mathcal{I} + (1 + \frac{3}{\delta})^p \mathtt{cost}_\mathcal{I}(\mathcal{T})\right) \leq \beta \leq \frac{2}{mt}\left(2\mathtt{opt}_\mathcal{I} + (1 + \frac{3}{\delta})^p \mathtt{cost}_\mathcal{I}(\mathcal{T})\right)$. *Then Algorithm 3 finds a set* $\mathcal{S} \in \mathcal{F}$ *in time* $n^{m^2 \cdot 2^{O(p)}}$ *such that* $|\mathcal{S}| \leq k$ *and* $\mathtt{cost}_\mathcal{I}(\mathcal{S}) \leq (O(1) + (1 + \epsilon')^p)\left(\mathtt{cost}_\mathcal{I}(\mathcal{T}) + \mathtt{opt}_\mathcal{I}\right)$.

## 5   EMPIRICAL STUDY

We compare our algorithm with previously best algorithms in the literature on benchmark datasets for socially fair $k$-median problem. Namely, we compare our bicriteria algorithm with Abbasi et al. (2021) (ABV), and our exact algorithm (that outputs *exactly* $k$ centers) with Makarychev and Vakilian (2021) (MV). Since our bicriteria algorithm produces only a small number of extra centers (e.g., for two groups, our algorithm only produces one extra center — see Section E.3), we search over the best $k$-subset in the set of $k + m$ selected centers. However, instead of performing an exhaustive search combinatorially, we use a mixed-integer linear programming solver to find the best $k$-subset. Our code is written in MATLAB. We use IBM ILOG CPLEX 12.10 to solve the linear programs (and mixed-integer programs). We used a MacBook Pro (2019) with a 2.3 GHz 8-Core Intel Core i9 processor, a 16 GB 2667 MHz DDR4 memory card, a Intel UHD Graphics 630 1536 MB graphic card, 1 TB of SSD storage, and macOS version 12.3.1.

**Datasets.** We use three benchmark datasets that have been extensively used in the fairness literature. Similar to other works in fair clustering Chierichetti et al. (2017), we subsample the points in the datasets. Namely, we consider the first 500 examples in each dataset. A quick overview of the used datasets is in the following. (1) **Credit dataset** Yeh and Lien (2009) consists of records of 30000 individuals with 21 features. We divided the multi-categorical education attribute to two demographic groups: "higher educated" and "lower educated." (2) **Adult dataset** Kohavi et al. (1996); Asuncion and Newman (2007) contains records of 48842 individuals collected from census data, with 103 features. We consider five racial groups of "Amer-Indian-Eskim", "AsianPac-Islander", "Black", "White", and "Other" for one of our experiments. For another experiment we consider the *intersectional* groups of race and gender (male and female) that results in 10 groups. (3) **COMPAS dataset** Angwin et al. (2016) is gathered by ProPublica and contains the recidivism rates for 9898 defendants. The data is divided to two racial groups of African-Americans (AA) and Caucasians (C). The results for the COMPAS datasets are included in the appendix.

**Bicriteria approximation.** The ABV algorithm, first solves the natural LP relaxation and then uses the "filtering" technique Lin and Vitter (1992); Charikar et al. (2002) to round the fractional solution to an integral one. Given a parameter $0 < \epsilon < 1$, the algorithm outputs at most $k/(1 - \epsilon)$ centers and guarantees a $2/\epsilon$ approximation. In our comparison, we consider $\epsilon$ that gives almost the same number of centers as our algorithm. Tables in Section E.3, summarise the number of selected centers for different $k$ and $\epsilon$. The $\lambda$ parameter in our algorithm (see Algorithm 1 and Lemma 2.1) determines the factor of decrease in the radii of client balls in the iterative rounding algorithm. As illustrated in

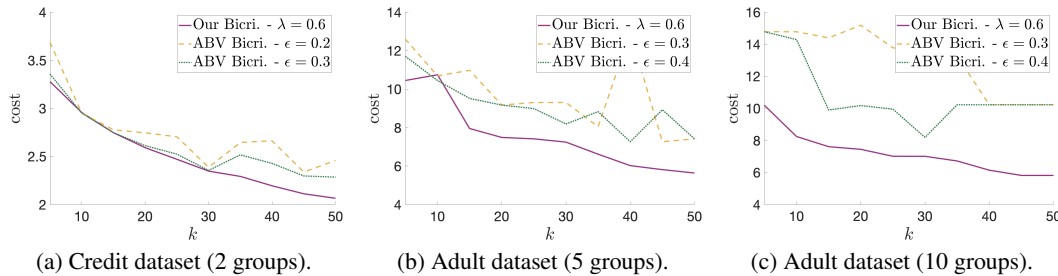

(a) Credit dataset (2 groups).   (b) Adult dataset (5 groups).   (c) Adult dataset (10 groups).

Figure 2: Comparison of our bicriteria algorithm with ABV Abbasi et al. (2021). The number of centers our algorithm selects is close to $k$ and is often smaller than ABV (see Section E.3).

Section E.2, the performance of our algorithm do not change significantly by changing $\lambda$. So in our comparisons, we fix $\lambda = 0.6$. Figure 4 illustrates that our algorithm outperforms ABV on different benchmark datasets. The gap between the performance of our algorithm and ABV becomes larger as the number of groups and $k$ become larger. For example, for the Adult dataset with 10 groups and $k = 50$, the objective value of ABV is almost twice of the objective that our algorithm achieves.

**Exactly $k$ centers.** The MV algorithm, fist sparsifies the linear programming relaxation by setting the connection variables of points that are far from each other to zero. It then adopts a randomized rounding algorithm similar to Charikar et al. (2002) based on consolidating centers and points. In the process of rounding, it produces a $(1 - \gamma)$-restricted solution which is a solution where each center is either open by a fraction of at least $(1 - \gamma)$ or it is zero. The algorithm needs $\gamma < 0.5$. The results of MV for different values of $\gamma$ are presented in Section E.2. It appears that MV performs better for larger values of $\gamma$, so below we use $\gamma = 0.1$ and $\gamma = 0.4$ for our comparisons. Figure 5 illustrates that our algorithms outperforms MV on different benchmark datasets. Similar to the bicriteria case, the gap between the performance of our algorithm and MV becomes larger as the number of groups and $k$ become larger. For example, for the Adult dataset with 5 or 10 groups and $k = 50$, the objective value of MV is almost thrice of the objective that our algorithm achieves.

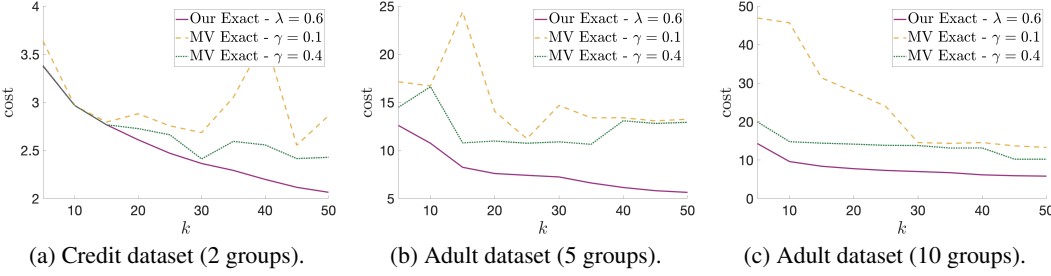

(a) Credit dataset (2 groups).   (b) Adult dataset (5 groups).   (c) Adult dataset (10 groups).

Figure 3: Comparison of our algorithm with $k$ centers with MV Makarychev and Vakilian (2021).

**Empirical running time.** We compare the running time of our algorithms with that of ABV and MV on the three datasets (see Section E.4). To summarize, the running times of our bicriteria algorithm and the exact algorithm with exhaustive search are virtually the same when the number of groups is no more than 5. Moreover our algorithms' times are comparable to ABV and are significantly less than MV in most cases. The latter is because MV needs to guess the value of the optimal objective value. Therefore it needs to run the algorithm multiple times. We run the algorithm with 5 different values (by multiplying different factors of two) and output the best out of these for MV.

## 6 CONCLUSION

We presented a polynomial time bicrteria algorithm for the socially fair $(\ell_p, k)$-clustering with $m$ groups that outputs at most $k + m$ centers. Using this, we presented two different constant-factor approximation algorithms that return exactly $k$ centers. An interesting future work is to investigate the use of recently introduced techniques in $k$-means++ Lattanzi and Sohler (2019); Choo et al. (2020) forw faster constant-factor approximation algorithms for socially fair $k$-means. It is also interesting to explore scalable algorithms, for example, using the coreset framework.

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

## A  EXHAUSTIVE SEARCH

**Lemma A.1** (Goyal and Jaiswal (2021)). *Let $k' > k$ and $S$ be a set of centers of size $k'$ and cost $C$ for the socially fair $(\ell_p, k)$-clustering problem with $m$ groups. Let $T \subset S$ be a set of size $k$ with minimum cost among all subsets of size $k$ of $S$. Then the cost of $T$ is less than or equal to $3^{p-1}(C + 2\text{opt})$ where $\text{opt}$ is the cost of the optimal solution.*

*Proof.* Let $\text{OPT}$ be an optimal set of centers. For each center $o \in \text{OPT}$, let $s_o$ be the closest center in $S$ to $o$, i.e., $s_o := \arg\min_{s \in S} d(s, o)$. Let $T' := \{s_o : o \in \text{OPT}\}$. Because the size of $\text{OPT}$ is $k$, $|T'| \leq k$. We show that the cost of $T'$ is less than or equal to $3^{p-1}(C + 2\text{opt})$. The result follows from this because $T' \subset S$ and $|T'| \leq k$.

Let $i$ be a client and $o_i$ be the closest facility in $\text{OPT}$ to $i$. Let $t'_i$ be the closest facility to $o_i$ in $T'$ which means $t'_i$ is also the closest facility to $o_i$ in $S$. Moreover let $s_i$ be the closest facility to $i$ in $S$. By triangle inequality, $d(i, t'_i) \leq d(i, o_i) + d(o_i, t'_i)$. By definition of $t'_i$, $d(o_i, t'_i) \leq d(o_i, s_i)$. Therefore $d(i, t'_i) \leq d(i, o_i) + d(o_i, s_i)$. Moreover by triangle inequality $d(o_i, s_i) \leq d(i, o_i) + d(i, s_i)$. Therefore $d(i, t'_i) \leq 2d(i, o_i) + d(i, s_i)$. Taking both sides to the power of $p$ and using the power mean inequality, i.e., $(x + y + z)^p \leq 3^{p-1}(x^p + y^p + z^p)$, we conclude $d(i, t'_i)^p \leq 3^{p-1}(2d(i, o_i)^p + d(i, s_i)^p)$. The result follows from summing such inequality for each group and taking the maximum over groups. $\square$

## B  OMITTED PROOFS OF SECTION 2

*Proof of Claim 2.2.* In each iteration, we put at most one client in $\mathcal{U}^\star$. For this client, Eq. 8 is tight, i.e., $\sum_{j \in B_i} y_j = 1$. Note that we update $F_i$ to $B_i$. Therefore the new point in $\mathcal{U}^\star$ satisfies Eq. 7. Moreover, for a point $i'$ that is removed from $\mathcal{U}^\star$, we have

$$\sum_{j \in B_{i'}} y_j \leq \sum_{j \in F_{i'}} y_j = 1.$$

Therefore such a point satisfies Eq. 8. Hence a feasible solution to the LP of iteration $t$ is also feasible for iteration $t + 1$. Therefore the cost of the LP is non-increasing over iterations.

The second statement follows since if no constraint from Eq. 8 is tight, then the linear program is the intersection of a matroid polytope with $m$ linear constraints and the result follows from Lemma 1.3. $\square$

*Proof of Claim 2.3.* Let $t$ be the iteration where $D_{i'}$ is updated for the last time. If $D_{i'}$ is only set once at Line 4 of Algorithm 1 and it is never updated, then $t = 0$. We first show that immediately after iteration $t$, there is one total facility at a distance of at most $3D_{i'}$ from $i'$. If $t = 0$, then there existed $i \in \mathcal{U}^\star$ such that $F_i \cap F_{i'} \neq \emptyset$ and $D_i \leq D_{i'}$. Therefore by triangle inequality, all the facilities in $F_i$ are within a distance of at most $3D_{i'}$ from $i'$, see Figure 1 (a). Hence because Eq. 7 enforces one total facility in $F_i$, there exists one total facility at a distance of at most $3D_{i'}$ from $i'$. If $t > 0$, then $i'$ is moved from $\mathcal{U}^\star$ to $\mathcal{U}^f$ because at iteration $t$, a facility $i$ is added to $\mathcal{U}^\star$ such that $D_i < D_{i'}$ and $F_i \cap F_{i'} \neq \emptyset$ — see the condition of Procedure Update-$\mathcal{U}^\star(i)$ in Algorithm 1. Again because of enforcement of Eq. 7 and triangle inequality, there exists one total facility at a distance of at most $3D_{i'}$ from $i'$ immediately after iteration $t$.

Now note that after iteration $t$, the facility $i \in \mathcal{U}^\star$ with $F_i \cap F_{i'} \neq \emptyset$ and $D_i \leq D_{i'}$ might get removed from $\mathcal{U}^\star$. In which case, we do not have the guarantee of Eq. 7 any longer. Let $i_0 := i$. We define $i_{p+1}$ to be the client that has caused the removal of client $i_p$ (through Procedure Update-$\mathcal{U}^\star(i)$) from $\mathcal{U}^\star$ after iteration $t$. Note that by the condition of Update-$\mathcal{U}^\star(i)$) from $\mathcal{U}^\star$, $D_{i_{p+1}} < D_{i_p}$. Therefore because we have rounded the distances to multiples of $(1 + \lambda)$, we have $D_{i_{p+1}} \leq \frac{D_{i_p}}{1+\lambda}$. Let $i_r$ be the last point in this chain, i.e., $i_r$ has caused the removal of $i_{r-1}$ and $i_r$ has stayed in $\mathcal{U}^\star$ until termination of the algorithm. Then by guarantee of Eq. 7 and triangle inequality, there is one total facility for $i'$ within a distance of

$$D_{i'} + \sum_{j=0}^{r} 2D_{i_j} \leq D_{i'} + 2\sum_{j=0}^{r} \frac{D_{i'}}{(1+\lambda)^j} \leq \left(1 + \frac{2(1+\lambda)}{\lambda}\right) D_{i'}.$$

□

*Proof of Claim 2.4.* By Claim 2.2, at every iteration, the cost of the linear program only decreases since a feasible solution to previous iteration remains feasible for the next iteration. Thus the objective value of $\hat{y}$ in $\text{LP}(\mathcal{U}^\star, \mathcal{U}^f, D)$ is at most $(1 + \lambda)^p$ the optimal cost of **LP1** (where we lost the factor of $(1 + \lambda)^p$ by rounding all distances to powers of $(1 + \lambda)$).

We now construct $\hat{x}$ such that $(\hat{x}, \hat{y})$ is feasible to **LP1**. First note that the above procedure always terminates. We construct $\hat{x}$ by processing clients one by one. We process the clients in $\mathcal{U}^f$ and $\mathcal{U}^\star$ as follows. For any $i \in \mathcal{U}^\star$, we define $\hat{x}_{ij} = \hat{y}_j$ for each $j \in F_i$. Observe that we have $\sum_{j \in F_i} x_{ij} = 1$ for such $i \in \mathcal{U}^\star$ and we obtain feasibility for this client. For any $i \in \mathcal{U}^f$, we define $\hat{x}_{ij} = \hat{y}_j$ for each $j \in B_i$. Observe that we only insisted $\sum_{j \in B_i} \hat{y}_j \leq 1$ and therefore we still need to find $1 - \sum_{j \in B_i} \hat{x}_{ij} = 1 - \sum_{j \in B_i} \hat{y}_j$ facilities to assign to client $i$. For this remaining amount $1 - \hat{y}(B_i)$, we notice by Claim 2.3, there is at least one facility within distance $\left(1 + \frac{2(1+\lambda)}{\lambda}\right) D_i$ of this client. Thus we can assign the remaining $1 - \hat{y}(B_i)$ facility to client $i$ at a distance of no more than $\left(1 + \frac{2(1+\lambda)}{\lambda}\right) D_i$. Note that the cost is only increased by a factor of $\left(1 + \frac{2(1+\lambda)}{\lambda}\right)^p$. □

## C  OMITTED PROOFS OF SECTION 3

*Proof of Lemma 3.2.* First note that a facility $i$ in $\text{OPT}_\mathcal{I}$ cannot be $\alpha$-dense because $d(i, \text{OPT}_\mathcal{I})) = 0$. Let $(j_1, j_1'), \ldots, (j_\ell, j_\ell')$ be a sequence of pairs of facilities such that for every $b = 1, \ldots, \ell$,

- $j_b \in \mathcal{F} \setminus \bigcup_{z=1}^{b-1} \text{FBall}_\mathcal{I}(j_z, d(j_z, j_z'))$ is an $\frac{\text{opt}_\mathcal{I}}{mt}$-dense facility; and

- $j_b'$ is the closest facility to $j_b$ in $\text{OPT}_\mathcal{I}$.

We show that $\ell \leq m^2 t$. For $b \in [\ell]$ and $s \in [m]$, let $\mathcal{B}_{b,s} := \text{CBall}_{\mathcal{I},s}(j_b, \frac{1}{3}d(j_b, j_b'))$. First we show that for any group $s \in [m]$, the client balls $\mathcal{B}_{1,s}, \ldots, \mathcal{B}_{\ell,s}$ are disjoint. Let $1 \leq z < w \leq \ell$. By triangle inequality $d(j_w, j_z') \leq d(j_w, j_z) + d(j_z, j_z')$. Moreover by definition $j_w \notin \text{FBall}_\mathcal{I}(j_z, d(j_z, j_z'))$. Thus $d(j_z, j_z') \leq d(j_w, j_z)$. Hence $d(j_w, j_z') \leq 2d(j_w, j_z)$. Since $j_w'$ is the closest facility to $j_w$ in $\text{OPT}_\mathcal{I}$, $d(j_w, j_w') \leq d(j_w, j_z')$. Therefore $d(j_w, j_w') \leq 2d(j_w, j_z)$. Combining this with $d(j_z, j_z') \leq d(j_w, j_z)$ implies $\frac{1}{3}(d(j_z, j_z') + d(j_w, j_w')) \leq d(j_z, j_w)$. If $\mathcal{B}_{z,s}$ and $\mathcal{B}_{w,s}$ overlap then there exists $u \in \mathcal{B}_{z,s} \cap \mathcal{B}_{w,s}$ and by triangle inequality, $d(j_z, j_w) \leq d(j_z, u) + d(j_w, u) < \frac{1}{3}d(j_z, j_z') + \frac{1}{3}d(j_w, j_w')$, which is a contradiction.

Therefore for $s \in [m]$, $\mathcal{B}_{1,j}, \ldots, \mathcal{B}_{\ell,j}$ are disjoint. Also since $A_1, \ldots, A_s$ are disjoint, all of $\mathcal{B}_{b,s}$'s are disjoint for $b \in [\ell]$ and $s \in [m]$. By definition, for any $b \in [\ell]$, there exists $s_b \in [m]$ such that $\left(\frac{2}{3}d(j_b, \text{OPT}_\mathcal{I})\right)^p |\mathcal{B}_{b,s_b}| > \frac{\text{opt}_\mathcal{I}}{mt}$. Therefore, if $\ell > m^2 t$, $\sum_{b=1}^\ell \left(\frac{2}{3}d(j_b, \text{OPT}_\mathcal{I})\right)^p |\mathcal{B}_{b,s_b}| > m\text{opt}_\mathcal{I}$. Thus

$$m \cdot \max_{s \in [m]} \sum_{b=1}^\ell \left(\frac{2}{3}d(j_b, \text{OPT}_\mathcal{I})\right)^p |\mathcal{B}_{b,s}| \geq \sum_{s \in [m]} \sum_{b=1}^\ell \left(\frac{2}{3}d(j_b, \text{OPT}_\mathcal{I})\right)^p |\mathcal{B}_{b,s}| > m\text{opt}_\mathcal{I}.$$

Note that the connection cost of a client in $\mathcal{B}_{b,s}$ in the optimal solution is at least $\left(\frac{2}{3}d(j_b, \text{OPT}_\mathcal{I})\right)^p = \left(\frac{2}{3}d(j_b, j_b')\right)^p$. Therefore, as the $\mathcal{B}_{b,s}$'s are disjoint, $\text{opt}_\mathcal{I} \geq \max_{s \in [m]} \sum_{b=1}^\ell \left(\frac{2}{3}d(j_b, \text{OPT}_\mathcal{I})\right)^p |\mathcal{B}_{b,s}|$. This is a contradiction. Therefore $\ell \leq m^2 t$. Thus Algorithm 2 returns an instance with the desired properties. □

## D  OMITTED PROOFS OF SECTION 4

*Proof of Lemma 4.2.* If Algorithm 3 ends in Step 7, then $\text{cost}_\mathcal{I}(\mathcal{T}) + m\beta$ is at most $\text{cost}_\mathcal{I}(\mathcal{T}) + \frac{2}{t}(2\text{opt}_\mathcal{I} + (1 + \frac{3}{\delta})^p \text{cost}_\mathcal{I}(\mathcal{T})) = O(\text{opt}_\mathcal{I} + \text{cost}_\mathcal{I}(\mathcal{T}))$. Otherwise, we run the loop. Now we show that there exist sets $\mathcal{D}_0 \subseteq \mathcal{T}'$ and $\mathcal{V}_0 \subseteq \mathcal{F}$ such that $|\mathcal{V}_0| < m^2 t$, $|\mathcal{D}_0| + |\mathcal{V}_0| = k$, and $\mathcal{S}_{\mathcal{D}_0, \mathcal{V}_0}$ satisfies the desired properties. For a facility $j \in \mathcal{T}'$, let $L_j = d(j, \mathcal{T}' \setminus \{j\})$ and

$\ell_j = d(j, \text{OPT}_{\mathcal{I}})$. We say $j \in \mathcal{I}$ is *determined* if $\ell_j \leq \delta L_j$. Otherwise, we say $j$ is *undetermined*. Let $\mathcal{D}_0 = \{j \in \mathcal{T}' : \ell_j \leq \delta L_j\}$. For $j \in \mathcal{D}_0$, let $f_j^*$ be the closest facility to $j$ in $\text{OPT}_{\mathcal{I}}$. Let $\mathcal{V}_0 = \text{OPT}_{\mathcal{I}} \setminus \{f_j^* : j \in \mathcal{D}_0\}$. First note that for any two distinct facilities in $j, j' \in \mathcal{D}_0$, $d(j, j') \geq \max\{L_j, L_{j'}\}$. Moreover by definition, $d(j, f_j^*) \leq \delta L_j \leq \delta \max\{L_j, L_{j'}\}$. Therefore by triangle inequality, $d(j', f_j^*) \geq (1 - \delta) \max\{L_j, L_{j'}\}$. Moreover by definition and because $\delta \in (0, \frac{1}{8})$, $(1 - \delta) \max\{L_j, L_{j'}\} > \delta L_{j'} \geq d(j', f_{j'}^*)$. Therefore $d(j', f_j^*) > d(j', f_{j'}^*)$. Thus for any two distinct $j, j' \in \mathcal{D}_0$, $f_j^* \neq f_{j'}^*$.

Therefore $|\{f_j^* : j \in \mathcal{D}_0\}| = |\mathcal{D}_0|$. Thus $|\mathcal{V}_0| = |\text{OPT}_{\mathcal{I}}| - |\mathcal{D}_0| = k - |\mathcal{D}_0|$. Let $\mathcal{U}_0 = \mathcal{T}' \setminus \mathcal{D}_0$ be the set of undetermined facilities. Since $|\mathcal{T}'| > k$, $|\mathcal{V}_0| = k - |\mathcal{D}_0| = k - |\mathcal{T}'| + |\mathcal{U}_0| < |\mathcal{U}_0|$. We show $|\mathcal{U}_0| < m^2 t$. For every $j \in \mathcal{T}'$ and $s \in [m]$, let $A_{s,j}$ be the set of clients of group $s$ that are connected to $j$ in solution $\mathcal{T}'$ and let $C_{s,j}$ be the total connection cost of these clients. Therefore $\text{cost}_{\mathcal{I}}(\mathcal{T}') = \max_{s \in [m]} \sum_{j \in \mathcal{T}'} C_{s,j}$. Let $j^* := \arg\min_{j \in \mathcal{U}_0} \sum_{s \in [m]} C_{s,j}$. Let $\tilde{j}$ be the closest facility to $j^*$ in $\mathcal{T}' \setminus \{j^*\}$, i.e., $d(j^*, \tilde{j}) = L_{j^*}$. Then $\text{cost}_{\mathcal{I}}(\mathcal{T}' \setminus \{j^*\}) - \text{cost}_{\mathcal{I}}(\mathcal{T}') \leq \max_{s \in [m]} \sum_{i \in A_{s,j^*}} d(i, \tilde{j})^p$. For $s \in [m]$, let $A_{s,j^*}^{\text{in}} := A_{s,j^*} \cap \text{CBall}_{\mathcal{I},s}(j^*, \frac{1}{3}\delta L_{j^*})$ and $A_{s,j^*}^{\text{out}} := A_{s,j^*} \setminus A_{s,j^*}^{\text{in}}$. By triangle inequality, for any $i \in A_{s,j^*}^{\text{in}}$, $d(i, \tilde{j}) \leq (1 + \frac{1}{3}\delta) L_{j^*}$. Moreover since $j^*$ is undetermined, $d(i, \tilde{j}) < (1 + \frac{1}{3}\delta)\frac{1}{\delta}\ell_{j^*} = (\frac{1}{\delta} + \frac{1}{3})\ell_{j^*} < \frac{2}{3}\ell_{j^*}$, and for any $s \in [m]$, $\text{CBall}_{\mathcal{I},s}(j^*, \frac{1}{3}\delta L_{j^*}) \subseteq \text{CBall}_{\mathcal{I},s}(j^*, \frac{1}{3}\ell_{j^*})$. Thus $\sum_{i \in A_{s,j^*}^{\text{in}}} d(i, \tilde{j})^p < (\frac{2}{3}\ell_{j^*})^p |\text{CBall}_{\mathcal{I},s}(j^*, \frac{1}{3}\ell_{j^*})|$. Therefore since $\mathcal{I}$ is a $\frac{\text{opt}_{\mathcal{I}}}{mt}$-sparse instance, $\sum_{i \in A_{s,j^*}^{\text{in}}} d(i, \tilde{j})^p \leq \frac{\text{opt}_{\mathcal{I}}}{mt}$. For $i \in A_{s,j^*}^{\text{out}}$, $d(i, j^*) \geq \frac{1}{3}\delta L_{j^*}$. Thus $\frac{3}{\delta} d(i, j^*) \geq L_{j^*}$ and by triangle inequality, $d(i, \tilde{j}) \leq d(i, j^*) + d(j^*, \tilde{j}) = d(i, j^*) + L_{j^*} \leq (1 + \frac{3}{\delta}) d(i, j^*)$. Therefore

$$\text{cost}_{\mathcal{I}}(\mathcal{T}' \setminus \{j^*\}) - \text{cost}_{\mathcal{I}}(\mathcal{T}') \leq \max_{s \in [m]} \left( \frac{\text{opt}_{\mathcal{I}}}{mt} + (1 + \frac{3}{\delta})^p C_{s,j^*} \right)$$
$$\leq \frac{\text{opt}_{\mathcal{I}}}{mt} + (1 + \frac{3}{\delta})^p \sum_{s \in [m]} C_{s,j^*}. \tag{10}$$

By definition, $\sum_{s \in [m]} C_{s,j^*} = \min_{j \in \mathcal{U}_0} \sum_{s \in [m]} C_{s,j} \leq \frac{m \text{cost}_{\mathcal{I}}(\mathcal{T}')}{|\mathcal{U}_0|}$. So if $|\mathcal{U}_0| \geq m^2 t$, then $\sum_{s \in [m]} C_{s,j^*} \leq \frac{\text{cost}_{\mathcal{I}}(\mathcal{T}')}{mt}$. Moreover, since $|\mathcal{T} \setminus \mathcal{T}'| < m$,

$$\text{cost}_{\mathcal{I}}(\mathcal{T}') < \text{cost}_{\mathcal{I}}(\mathcal{T}) + m\beta \leq \text{cost}_{\mathcal{I}}(\mathcal{T}) + \frac{2}{t}\left(2\text{opt}_{\mathcal{I}} + (1 + \frac{3}{\delta})^p \text{cost}_{\mathcal{I}}(\mathcal{T})\right)$$
$$\leq \left(\frac{1}{1 + \frac{3}{\delta}}\right)^p \text{opt}_{\mathcal{I}} + \frac{3}{2}\text{cost}_{\mathcal{I}}(\mathcal{T}).$$

Combining with Eq. 10, $\text{cost}_{\mathcal{I}}(\mathcal{T}' \setminus \{j^*\}) - \text{cost}_{\mathcal{I}}(\mathcal{T}') \leq 2\frac{\text{opt}_{\mathcal{I}}}{mt} + \frac{3}{2}(1 + \frac{3}{\delta})^p \frac{\text{cost}_{\mathcal{I}}(\mathcal{T})}{mt} \leq \beta$. This is a contradiction because $j^*$ should be removed in Step 4 of Algorithm 3. Therefore $|\mathcal{U}_0| < m^2 t$.

Now, we need to bound the cost of $\mathcal{S}_{\mathcal{D}_0, \mathcal{V}_0}$. For $j \in \mathcal{D}_0$ and $s \in [m]$, let $i \in \text{CBall}_{\mathcal{I},s}(j, \frac{1}{3}L_j)$. By triangle inequality the distance of $i$ to any facility in $\text{FBall}_{\mathcal{I}}(j, \delta L_j)$ is at most $(\frac{1}{3} + \delta)L_j$. For a facility $j' \in \mathcal{D}_0$, $j' \neq j$, by triangle inequality and because $d(j, j') \geq \max\{L_j, L_{j'}\}$, the distance of $i$ to any facility in $\text{FBall}_{\mathcal{I}}(j', \delta L_{j'})$ is at least

$$d(j, j') - \frac{L_j}{3} - \delta L_{j'} \geq d(j, j') - (\frac{1}{3} + \delta)d(j, j') = (\frac{2}{3} - \delta)d(j, j') \geq (\frac{2}{3} - \delta)L_j.$$

For $\delta < \frac{1}{8}$, we have $\frac{1}{3} + \delta < \frac{2}{3} - \delta$. Therefore, $i$ is either connected to $f_j$ or to a facility in $\mathcal{V}_0$. Let $\alpha_{f_j}^{(s,j)}$'s be as defined in Algorithm 3 for $\mathcal{D}_0$ and $\mathcal{V}_0$. Let $(\tilde{f}_j : j \in \mathcal{D}_0)$ be a $(1 + \epsilon)$-approximate solution for the following, obtained by Lemma 4.1. For $s \in [m]$, let $T_s = \bigcup_{j \in \mathcal{D}_0} \text{CBall}_{\mathcal{I},s}(j, L_j/3)$. Since for $j \in \mathcal{D}_0$, $f_j^* \in \text{OPT}_{\mathcal{I}}$'s are also in balls $\text{FBall}_{\mathcal{I}}(j, \delta L_j)$,

$$\max_{s \in [m]} \sum_{i \in T_s} d(i, \mathcal{S}_{\mathcal{D}_0, \mathcal{V}_0})^p \leq (1 + \epsilon) \max_{s \in [m]} \sum_{i \in T_s} d(i, \text{OPT}_{\mathcal{I}})^p.$$

Now consider a client $i \in A_s \setminus T_s$. If in the optimal solution, $i$ is connected to a facility in $\mathcal{V}_0$, then by definition, $d(i, \text{OPT}_{\mathcal{I}}) \geq d(i, \mathcal{S}_{\mathcal{D}_0, \mathcal{V}_0})$. Otherwise, in the optimal solution, $i$ is connected

to $f_j^* \in \mathtt{FBall}_{\mathcal{I}}(j, \delta L_j)$ for some $j \in \mathcal{D}_0$. We compare $d(i, \tilde{f}_j)$ to $d(i, f_j^*)$. Since $\tilde{f}_j, f_j^* \in \mathtt{FBall}_{\mathcal{I}}(j, \delta L_j)$, by triangle inequality and because $d(i, j) \geq \frac{L_j}{3}$, $\frac{d(i, \tilde{f}_j)^p}{d(i, f_j^*)^p} \leq \frac{(d(i,j)+\delta L_j)^p}{(d(i,j)-\delta L_j)^p} \leq \frac{(L_j/3+\delta L_j)^p}{(L_j/3-\delta L_j)^p} = \left(\frac{1+3\delta}{1-3\delta}\right)^p$. Thus because $\delta \leq \frac{1}{8}$, $\frac{1+3\delta}{1-3\delta} \leq 1 + 12\delta$. Moreover since $\delta < \frac{\log(1+\epsilon')}{12}$, $\mathtt{cost}_{\mathcal{I}}(\mathcal{S}_{\mathcal{D}_0, \mathcal{V}_0}) \leq (1 + \epsilon')^p \cdot \mathtt{opt}_{\mathcal{I}}$. Finally note that the loop runs for $n^{O(m^2 t)}$ iterations because $|\mathcal{V}| < m^2 t$ and $|\mathcal{T} \setminus \mathcal{D}| \leq m^2 t + m$. Moreover by Lemma 4.1, each iteration runs in $(nk)^{O(m^2/\epsilon)}$ time. $\qquad\square$

# E OMITTED EMPIRICAL RESULTS

## E.1 COMPARISON OF ALGORITHMS SHOWING BOTH MAXIMUM AND MINIMUM

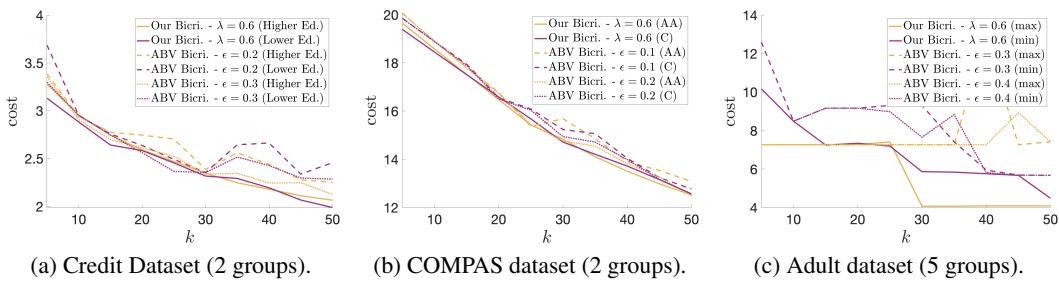

(a) Credit Dataset (2 groups).     (b) COMPAS dataset (2 groups).     (c) Adult dataset (5 groups).

Figure 4: Comparison of our bicriteria algorithm with ABV Abbasi et al. (2021). The max and min on Subfigure (c) are across the demographic groups and are used to prevent cluttering plots with 5 groups. The number of centers our algorithm selects is close to $k$ and is often smaller than ABV (see Section E.3).

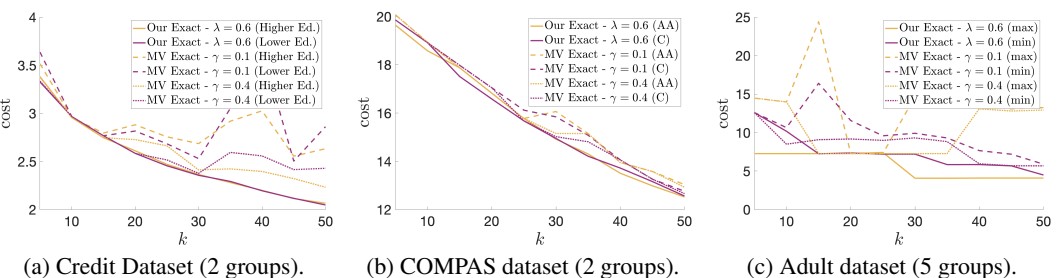

(a) Credit Dataset (2 groups).     (b) COMPAS dataset (2 groups).     (c) Adult dataset (5 groups).

Figure 5: Comparison of our algorithm with exactly $k$ centers with MV Makarychev and Vakilian (2021). The max and min on Subfigure (c) are across the groups and are used to prevent cluttering plots with 5 groups.

## E.2 RESULTS OF DIFFERENT ALGORITHMS FOR DIFFERENT PARAMETERS

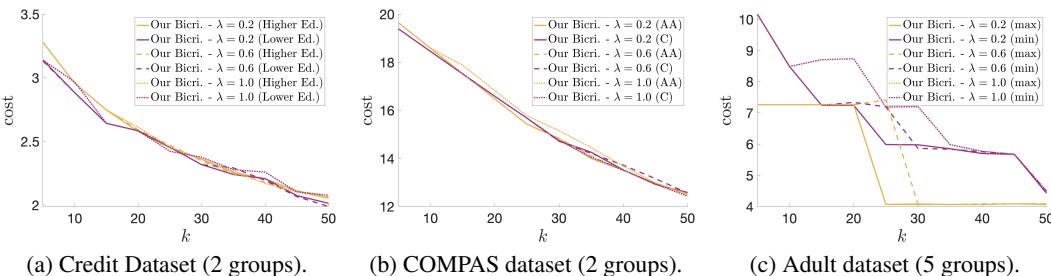

(a) Credit Dataset (2 groups).  (b) COMPAS dataset (2 groups).  (c) Adult dataset (5 groups).

Figure 6: Performance of our bicriteria algorithm for different values of $\lambda$. The max and min on Subfigure (c) are across the demographic groups.

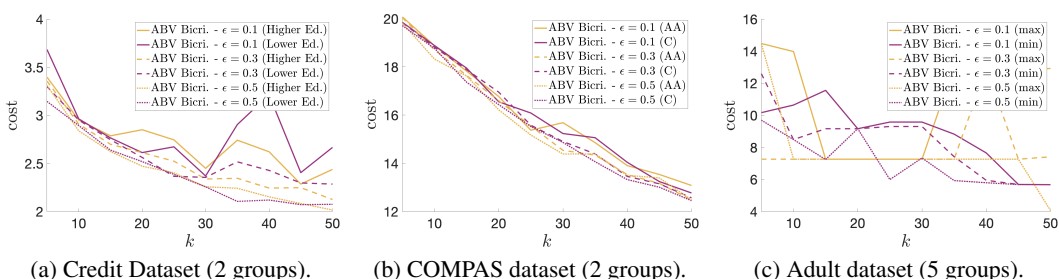

(a) Credit Dataset (2 groups).  (b) COMPAS dataset (2 groups).  (c) Adult dataset (5 groups).

Figure 7: Performance of our bicriteria algorithm of ABV Abbasi et al. (2021) for different values of $\epsilon$. The max and min on Subfigure (c) are across the demographic groups.

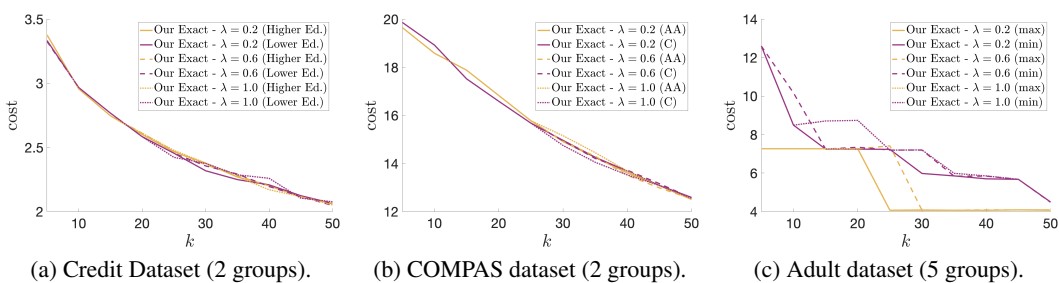

(a) Credit Dataset (2 groups).  (b) COMPAS dataset (2 groups).  (c) Adult dataset (5 groups).

Figure 8: Performance of our algorithm with exactly $k$ centers for different values of $\lambda$. The max and min on Subfigure (c) are across the demographic groups.

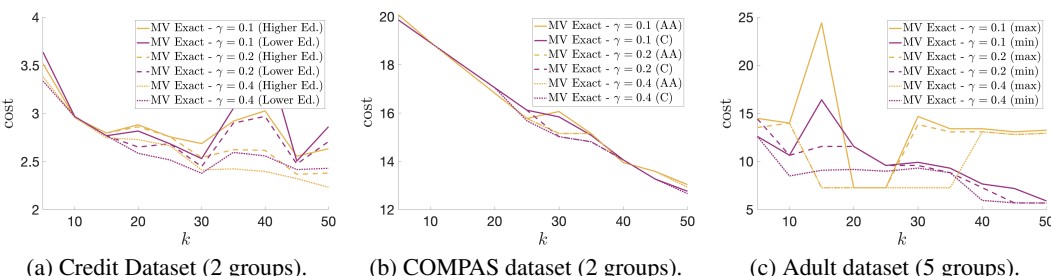

(a) Credit Dataset (2 groups).  (b) COMPAS dataset (2 groups).  (c) Adult dataset (5 groups).

Figure 9: Performance of our MV algorithm Makarychev and Vakilian (2021) for different values of $\gamma$. The max and min on Subfigure (c) are across the demographic groups.

### E.3 Number of Selected Centers in The Bicriteria Algorithms

Table 1: The number of selected centers for our bicriteria algorithm on the Credit dataset. $\lambda$ is a parameter of the algorithm and denotes the amount of decrease in radii of balls around the clients in the iterative rounding algorithm.

|  | $\lambda = 0.2$ | $\lambda = 0.4$ | $\lambda = 0.6$ | $\lambda = 0.8$ | $\lambda = 1.0$ |
|---|---|---|---|---|---|
| $k = 5$ | 6 | 6 | 6 | 6 | 6 |
| $k = 10$ | 11 | 11 | 11 | 10 | 11 |
| $k = 15$ | 16 | 16 | 16 | 15 | 15 |
| $k = 20$ | 21 | 21 | 21 | 20 | 20 |
| $k = 25$ | 26 | 26 | 26 | 25 | 25 |
| $k = 30$ | 31 | 31 | 31 | 30 | 30 |
| $k = 35$ | 36 | 36 | 36 | 35 | 35 |
| $k = 40$ | 41 | 40 | 40 | 40 | 40 |
| $k = 45$ | 46 | 45 | 45 | 46 | 46 |
| $k = 50$ | 50 | 50 | 50 | 51 | 51 |

Table 2: The number of selected centers for bicriteria algorithm of Abbasi-Bhaskara-Venkatasubramanian Abbasi et al. (2021) on the Credit dataset. $\epsilon$ is a parameter of the algorithm. The maximum number of selected centers is $k/(1 - \epsilon)$ which achieves a $2/\epsilon$ approximation factor.

|  | $\epsilon = 0.1$ | $\epsilon = 0.2$ | $\epsilon = 0.3$ | $\epsilon = 0.4$ | $\epsilon = 0.5$ |
|---|---|---|---|---|---|
| $k = 5$ | 5 | 6 | 7 | 8 | 10 |
| $k = 10$ | 11 | 12 | 14 | 16 | 20 |
| $k = 15$ | 16 | 18 | 21 | 23 | 29 |
| $k = 20$ | 20 | 23 | 26 | 29 | 34 |
| $k = 25$ | 22 | 27 | 31 | 35 | 39 |
| $k = 30$ | 29 | 34 | 37 | 41 | 46 |
| $k = 35$ | 25 | 29 | 36 | 40 | 45 |
| $k = 40$ | 26 | 30 | 38 | 43 | 48 |
| $k = 45$ | 36 | 40 | 46 | 53 | 59 |
| $k = 50$ | 38 | 45 | 51 | 58 | 64 |

Table 3: The number of selected centers for our bicriteria algorithm on the COMPAS dataset. $\lambda$ is a parameter of the algorithm and denotes the amount of decrease in radii of balls around the clients in the iterative rounding algorithm.

|  | $\lambda = 0.2$ | $\lambda = 0.4$ | $\lambda = 0.6$ | $\lambda = 0.8$ | $\lambda = 1.0$ |
|---|---|---|---|---|---|
| $k = 5$ | 6 | 6 | 6 | 6 | 6 |
| $k = 10$ | 11 | 10 | 11 | 11 | 10 |
| $k = 15$ | 16 | 16 | 16 | 16 | 16 |
| $k = 20$ | 21 | 21 | 21 | 20 | 20 |
| $k = 25$ | 26 | 26 | 25 | 26 | 26 |
| $k = 30$ | 31 | 31 | 31 | 31 | 31 |
| $k = 35$ | 36 | 36 | 36 | 36 | 36 |
| $k = 40$ | 40 | 41 | 41 | 41 | 40 |
| $k = 45$ | 46 | 46 | 46 | 46 | 46 |
| $k = 50$ | 51 | 51 | 51 | 51 | 50 |

Table 4: The number of selected centers for bicriteria algorithm of Abbasi-Bhaskara-Venkatasubramanian Abbasi et al. (2021) on the COMPAS dataset. $\epsilon$ is a parameter of the algorithm. The maximum number of selected centers is $k/(1 - \epsilon)$ which achieves a $2/\epsilon$ approximation factor.

|  | $\epsilon = 0.1$ | $\epsilon = 0.2$ | $\epsilon = 0.3$ | $\epsilon = 0.4$ | $\epsilon = 0.5$ |
|---|---|---|---|---|---|
| $k = 5$ | 5 | 6 | 7 | 8 | 10 |
| $k = 10$ | 11 | 12 | 14 | 15 | 16 |
| $k = 15$ | 16 | 18 | 19 | 20 | 21 |
| $k = 20$ | 22 | 24 | 24 | 25 | 27 |
| $k = 25$ | 27 | 29 | 30 | 31 | 32 |
| $k = 30$ | 30 | 34 | 35 | 36 | 38 |
| $k = 35$ | 35 | 40 | 40 | 42 | 43 |
| $k = 40$ | 42 | 46 | 48 | 49 | 51 |
| $k = 45$ | 45 | 50 | 51 | 52 | 53 |
| $k = 50$ | 50 | 56 | 58 | 59 | 61 |

Table 5: The number of selected centers for our bicriteria algorithm on the Adult dataset with 5 groups. $\lambda$ is a parameter of the algorithm and denotes the amount of decrease in radii of balls around the clients in the iterative rounding algorithm.

|  | $\lambda = 0.2$ | $\lambda = 0.4$ | $\lambda = 0.6$ | $\lambda = 0.8$ | $\lambda = 1.0$ |
|---|---|---|---|---|---|
| $k = 5$ | 7 | 7 | 7 | 5 | 7 |
| $k = 10$ | 15 | 14 | 12 | 14 | 12 |
| $k = 15$ | 18 | 15 | 18 | 14 | 19 |
| $k = 20$ | 24 | 20 | 22 | 18 | 20 |
| $k = 25$ | 30 | 27 | 27 | 25 | 28 |
| $k = 30$ | 34 | 34 | 33 | 30 | 33 |
| $k = 35$ | 38 | 38 | 38 | 35 | 38 |
| $k = 40$ | 43 | 43 | 43 | 40 | 44 |
| $k = 45$ | 45 | 47 | 47 | 45 | 46 |
| $k = 50$ | 54 | 54 | 54 | 54 | 50 |

Table 6: The number of selected centers for bicriteria algorithm of Abbasi-Bhaskara-Venkatasubramanian Abbasi et al. (2021) on the Adult dataset with 5 groups. $\epsilon$ is a parameter of the algorithm. The maximum number of selected centers is $k/(1 - \epsilon)$ which achieves a $2/\epsilon$ approximation factor.

|  | $\epsilon = 0.1$ | $\epsilon = 0.2$ | $\epsilon = 0.3$ | $\epsilon = 0.4$ | $\epsilon = 0.5$ |
|---|---|---|---|---|---|
| $k = 5$ | 4 | 5 | 6 | 7 | 7 |
| $k = 10$ | 6 | 7 | 10 | 11 | 12 |
| $k = 15$ | 9 | 10 | 11 | 12 | 13 |
| $k = 20$ | 13 | 13 | 14 | 15 | 17 |
| $k = 25$ | 15 | 17 | 19 | 20 | 22 |
| $k = 30$ | 18 | 19 | 21 | 23 | 28 |
| $k = 35$ | 22 | 25 | 29 | 32 | 37 |
| $k = 40$ | 30 | 37 | 39 | 45 | 50 |
| $k = 45$ | 45 | 51 | 57 | 61 | 68 |
| $k = 50$ | 45 | 50 | 56 | 61 | 68 |

Table 7: The number of selected centers for our bicriteria algorithm on the Adult dataset with 10 groups. $\lambda$ is a parameter of the algorithm and denotes the amount of decrease in radii of balls around the clients in the iterative rounding algorithm.

|          | $\lambda = 0.2$ | $\lambda = 0.4$ | $\lambda = 0.6$ | $\lambda = 0.8$ | $\lambda = 1.0$ |
|----------|------|------|------|------|------|
| $k = 5$  | 9    | 9    | 9    | 8    | 9    |
| $k = 10$ | 20   | 18   | 19   | 17   | 18   |
| $k = 15$ | 23   | 24   | 22   | 16   | 22   |
| $k = 20$ | 30   | 27   | 29   | 20   | 23   |
| $k = 25$ | 32   | 35   | 34   | 25   | 26   |
| $k = 30$ | 38   | 38   | 36   | 31   | 30   |
| $k = 35$ | 44   | 45   | 43   | 36   | 35   |
| $k = 40$ | 48   | 47   | 44   | 48   | 49   |
| $k = 45$ | 51   | 52   | 51   | 50   | 51   |
| $k = 50$ | 57   | 59   | 55   | 57   | 53   |

Table 8: The number of selected centers for bicriteria algorithm of Abbasi-Bhaskara-Venkatasubramanian Abbasi et al. (2021) on the Adult dataset with 10 groups. $\epsilon$ is a parameter of the algorithm. The maximum number of selected centers is $k/(1 - \epsilon)$ which achieves a $2/\epsilon$ approximation factor.

|          | $\epsilon = 0.1$ | $\epsilon = 0.2$ | $\epsilon = 0.3$ | $\epsilon = 0.4$ | $\epsilon = 0.5$ |
|----------|------|------|------|------|------|
| $k = 5$  | 2    | 5    | 6    | 6    | 6    |
| $k = 10$ | 5    | 6    | 8    | 9    | 10   |
| $k = 15$ | 7    | 8    | 9    | 11   | 12   |
| $k = 20$ | 11   | 12   | 14   | 15   | 16   |
| $k = 25$ | 13   | 15   | 16   | 18   | 22   |
| $k = 30$ | 18   | 20   | 23   | 28   | 31   |
| $k = 35$ | 22   | 26   | 29   | 33   | 38   |
| $k = 40$ | 26   | 35   | 40   | 42   | 46   |
| $k = 45$ | 40   | 46   | 53   | 56   | 61   |
| $k = 50$ | 39   | 48   | 55   | 62   | 69   |

### E.4 COMPARISON OF RUNNING TIME OF DIFFERENT ALGORITHMS IN PRACTICE

Table 9: Comparison of the running time of different algorithms on the first 200 samples of the Credit dataset averaged over five runs.

| k | Our Bicriteria ($\lambda=0.2$) | Our Bicriteria ($\lambda=0.6$) | ABV Bicriteria ($\epsilon=0.3$) | ABV Bicriteria ($\epsilon=0.4$) | Our Exact ($\lambda=0.2$) | Our Exact ($\lambda=0.6$) | MV Exact ($\gamma=0.1$) | MV Exact ($\gamma=0.2$) |
|---|---|---|---|---|---|---|---|---|
| 20 | 11.12 | 13.20 | 12.21 | 11.07 | 11.56 | 13.28 | 33.42 | 30.41 |
| 25 | 13.20 | 13.67 | 11.51 | 12.20 | 14.06 | 14.51 | 34.04 | 36.52 |
| 30 | 4.42 | 10.32 | 4.59 | 3.64 | 4.53 | 10.40 | 20.59 | 15.28 |
| 35 | 3.60 | 3.74 | 4.53 | 3.89 | 4.27 | 5.88 | 21.97 | 23.35 |
| 40 | 5.57 | 3.86 | 4.23 | 3.62 | 7.92 | 6.09 | 16.33 | 19.77 |

Table 10: Comparison of the running time of different algorithms on the first 200 samples of the COMPAS dataset averaged over five runs.

| k | Our Bicriteria ($\lambda=0.2$) | Our Bicriteria ($\lambda=0.6$) | ABV Bicriteria ($\epsilon=0.3$) | ABV Bicriteria ($\epsilon=0.4$) | Our Exact ($\lambda=0.2$) | Our Exact ($\lambda=0.6$) | MV Exact ($\gamma=0.1$) | MV Exact ($\gamma=0.2$) |
|---|---|---|---|---|---|---|---|---|
| 20 | 7.98 | 5.44 | 7.19 | 6.51 | 8.81 | 6.22 | 44.18 | 45.13 |
| 25 | 4.88 | 4.72 | 7.04 | 5.36 | 5.88 | 5.75 | 39.68 | 38.47 |
| 30 | 4.83 | 5.60 | 5.14 | 3.98 | 6.26 | 6.91 | 35.15 | 37.71 |
| 35 | 5.83 | 6.74 | 5.91 | 6.33 | 7.27 | 8.22 | 34.29 | 35.85 |
| 40 | 5.37 | 5.21 | 5.87 | 6.13 | 6.82 | 6.03 | 39.59 | 43.49 |

Table 11: Comparison of the running time of different algorithms on the first 200 samples of the Adult dataset with 5 race groups averaged over five runs.

| k | Our Bicriteria ($\lambda=0.2$) | Our Bicriteria ($\lambda=0.6$) | ABV Bicriteria ($\epsilon=0.3$) | ABV Bicriteria ($\epsilon=0.4$) | Our Exact ($\lambda=0.2$) | Our Exact ($\lambda=0.6$) | MV Exact ($\gamma=0.1$) | MV Exact ($\gamma=0.2$) |
|---|---|---|---|---|---|---|---|---|
| 20 | 13.31 | 12.88 | 11.96 | 11.28 | 17.01 | 19.69 | 73.09 | 75.06 |
| 25 | 9.86 | 11.35 | 10.74 | 12.54 | 16.73 | 18.3 | 54.87 | 55.06 |
| 30 | 7.36 | 8.36 | 8.24 | 9.10 | 15.04 | 14.85 | 44.91 | 47.48 |
| 35 | 6.91 | 8.09 | 6.10 | 6.84 | 11.74 | 14.45 | 40.51 | 36.48 |
| 40 | 5.76 | 4.12 | 5.47 | 4.69 | 8.12 | 6.42 | 24.69 | 23.54 |

Table 12: Comparison of the running time of different algorithms on the first 500 samples of the Adult dataset with 10 race and gender groups averaged over five runs.

| k | Our Bicriteria ($\lambda=0.2$) | Our Bicriteria ($\lambda=0.6$) | ABV Bicriteria ($\epsilon=0.3$) | ABV Bicriteria ($\epsilon=0.4$) | Our Exact ($\lambda=0.2$) | Our Exact ($\lambda=0.6$) | MV Exact ($\gamma=0.1$) | MV Exact ($\gamma=0.2$) |
|---|---|---|---|---|---|---|---|---|
| 20 | 166.2 | 159.7 | 136.5 | 125.5 | 414.3 | 212.5 | 597.0 | 686.2 |
| 25 | 171.7 | 195.0 | 165.3 | 149.2 | 250.3 | 254.4 | 753.3 | 679.2 |
| 30 | 185.1 | 150.3 | 152.6 | 163.5 | 335.3 | 153.5 | 690.1 | 690.5 |
| 35 | 176.5 | 176.3 | 178.4 | 175.9 | 265.4 | 183.3 | 786.8 | 795.5 |
| 40 | 168.7 | 170.2 | 157.6 | 181.4 | 223.6 | 199.7 | 867.8 | 816.6 |

