# OpenReview forum: "Constant-Factor Approximation Algorithms for Socially Fair $k$-Clustering"
_ICLR.cc/2023/Conference — Submitted to ICLR 2023_

### Official Review · Reviewer_8wgV · 2022-10-22

**Confidence:** 3
**Correctness:** 3
**Technical Novelty And Significance:** 3
**Empirical Novelty And Significance:** 2
**Recommendation:** 5

**Clarity, Quality, Novelty And Reproducibility:**

# Clarify
The context, key assumptions are overall well-discussed. The presentation is clear, and the use of figures and tables help the presentation a lot. However, I have some detailed comments:

1. In the second paragraph, "the facility location problem" is mentioned. I think there is a unique special problem called THE facility location problem, while what you discuss later is not this problem but clustering problems. Please change for clarify.

2. page 2, "factor-constant-factor" is a typo

3. Some of the phenomenons found in the experiment are not explained. For instance, why it is the case that your algorithm performs better when k is larger? I also observe in Section E.4 that the running time is also smaller when k is larger, which is counter-intuitive since even evaluating the cost of a center set for larger k requires more time. Also, in E.2, you mentioned that the performance of the algorithm is not very sensitive to \lambda - can you justify if this might be something likely to happen in general, or because of certain property of the data set?


# Quality
The technical part is sound, and the experiment seems to be comprehensive. The main weakness is the choice of baseline does not include naive algorithms, which is mentioned int he "strength and weakness" part.


# Originality

the techniques are mostly based on previous works, but the adoption to this paper seems nontrivial. The idea of considering small-m regime seems to be new, but I also find the assumption of small m not very well justified in the paper.

**Strength And Weaknesses:**

# Strength:

This small-m-O(1)-approx regime seems to be new, since previous results either violate the number of centers k by a multiplicative factor (which may be much larger than m when m is small), or is super-constant approximation. The results are obtained using LP-rounding techniques that were mostly developed in previous works for related problems, but I find the adoption of them nontrivial.

# Weakness:

The algorithms do not seem to scale on larger data sets, thus is not quite practical. For instance, even with only 200 - 500 samples, the running time is already tens - hundreds of seconds (as reported in Section E.4). Also, the current baselines are mostly previous approximation algorithms. What about considering some naive baselines? For instance, what if one uses CPLEX to solve the integer program exactly, is the running time indeed very much worse? For vanilla k-clustering, often sub-sampling based preprocessing can applied, so I’m wondering what about first doing a uniform sample of the points, and then apply all these algorithms only on the sample to obtain a set of centers?

**Summary Of The Paper:**

This paper studies approximation algorithms for the socially fair clustering problem. In the socially fair clustering problem, the input is a metric space, n data points, m groups of the n points whose union is n but may not be disjoint, and an integer k, the goal is to find a set of k center points C from the metric space, such that the maximum (\ell_p, k)-clustering cost among all groups, is minimized. Here, the (\ell_p, k)-clustering cost is the sum of p-th power of distances from every data point to a center C.

The paper focuses on the case when m is small. In particular, it gives two O(1)-approx algorithms, one runs in poly(nkm)-time but returns k + m centers, and the other returns exactly k centers but runs in (nk)^{poly(m)} time (or poly(n) k^m for a slightly worse constant ratio). Experiments have been conducted to compare the empirical performance of the proposed algorithms to recent works. The main observation is that the proposed algorithms generally have a better ratio, this is especially true when the number of clusters k is large.

**Summary Of The Review:**

The theory part of the paper is interesting, but I recommend a weak reject overall since the impractical nature of this algorithmic work makes it less relevant to the audience of ICLR.

---

> ### Author Response · Authors · 2022-11-19
> **Response to Reviewer 8wgV**
>
> > What about considering some naive baselines? For instance, what if one uses CPLEX to solve the integer program exactly, is the running time indeed very much worse? For vanilla k-clustering, often sub-sampling based preprocessing can applied, so I’m wondering what about first doing a uniform sample of the points, and then apply all these algorithms only on the sample to obtain a set of centers?
>
> Thanks for the suggestion. We performed some experiments to solve the integer program exactly with CPLEX. This appears to be much slower than all of the algorithms. For example, on the Adult dataset with 10 groups and $k=20$, the solver did not converge even after two hours. Although we believe, this approach is viable, we will add accurate running times for it in the final version of the paper. However, we used your suggestion to replace the combinatorial exhaustive search algorithm for finding the best $k$-subset of the selected $k+m$ centers, with an exact mixed-integer linear program solver. This significantly improved the running time of our algorithm, which returns exactly $k$ centers for the dataset with $10$ groups. With this new approach, our algorithm is significantly faster than the algorithm of Makarychev-Vakilian on all datasets. Please see the new running times on the rebuttal revision that we have submitted. Thanks for the great suggestion!
>
> > For instance, why it is the case that your algorithm performs better when k is larger? I also observe in Section E.4 that the running time is also smaller when k is larger, which is counter-intuitive since even evaluating the cost of a center set for larger k requires more time.
>
> Note that $m$ is a constant for our datasets. Moreover, one way of comparing the bicriteria algorithm of Abbasi-Bhaskara-Venkatasubramania with ours is to pick $\epsilon=m/k$ in their algorithm. In this case, both methods have a hardbound of $k+m$ centers. Then they give an approximation factor of $2k/m$ for the fair $k$-median problem, while our approximation factor is constant. Note that when $m$ is fixed, and $k$ is increasing, their approximation factor deteriorates. This can explain why our algorithm outperforms by a more significant margin for larger k.
>
> Regarding the running time, we would like to emphasize that most of the cost is from solving the original linear programming problem (LP1). It appears the solver we are using for this purpose (IBM ILOG CPLEX 12.10) solves this faster for larger $k$. This is perhaps due to the shape of the polytope for larger $k$ for which the solution converges faster.
>
> > Also, in E.2, you mentioned that the performance of the algorithm is not very sensitive to \lambda - can you justify if this might be something likely to happen in general, or because of certain property of the data set?
>
> The theoretical guarantee of the algorithm is formulated based on $\lambda$ in Claim 2.4. Unless $\lambda$ is very close to zero, it does not change the approximation factor significantly for the k-median problem. For example, the approximation factor achieved for $\lambda=0.5$ and $0.6$ are $10.5$ and $10.13$, respectively.
>
> > The technical part is sound, and the experiment seems to be comprehensive. The main weakness is the choice of baseline does not include naive algorithms, which is mentioned in the "strength and weakness" part.
>
> Please see the response to the question above. Our experiments indicate that solving the problem directly using an exact mixed-integer linear programming solver is much slower than our approach and other approaches in the literature.
>
> > the techniques are mostly based on previous works, but the adoption to this paper seems nontrivial. The idea of considering small-m regime seems to be new, but I also find the assumption of small m not very well justified in the paper.
>
> Note that $m$ determines the demographic groups in the socially fair clustering problem. In all datasets in the fairness literature, there are only a handful of demographic groups (e.g., racial and gender groups). So we expect $m$ to be far smaller than $k$ in most applications. We will add a discussion about this to the paper.

---

> > ### Comment · Reviewer_8wgV · 2022-11-24
> > **Thanks for the responses**
> >
> > > With this new approach, our algorithm is significantly faster than the algorithm of Makarychev-Vakilian on all datasets. Please see the new running times on the rebuttal revision that we have submitted. Thanks for the great suggestion!
> >
> > I'm glad to hear that the trick helps to improve the running time. However, after looking at the new version of the paper, I can't find a clear description of this. I'm not even sure which are the parts that have been changed.
> >
> > Another question: if one applies subsampling, it often achieves a worse accuracy. How does it affect the accuracy in your experiments? Perhaps an accuracy vs sample-size (and/or running time) comparison is worth conducting. (In fact, I didn't find how the sample size is picked in the revised version, but I might have missed it.)
> >
> > Finally, my main concern is still about the practical scalability of the algorithm. Even after applying the trick that I suggested, it still needs 5s - 10s to finish a 200-point instance as shown in the updated Sec E.4, which does not seem to scale easily to even a moderately large datasets (e.g., with 10k points).

---

> > > ### Author Response · Authors · 2022-11-24
> > > **Second Response to Reviewer 8wgV**
> > >
> > > Thank you for the second round of review!
> > >
> > > > However, after looking at the new version of the paper, I can't find a clear description of this.
> > >
> > > The description of the new approach is included in the first paragraph of Section 5. Essentially instead of doing an exhaustive search combinatorially to find the best $k$-subset out of the $k+m$ centers selected by our iterative rounding algorithm, we find the best $k$-subset by solving a mixed-integer linear program. Therefore the matrix of distances is $n$-by-$(k+m)$, and we solve (LP1) described in Section 2, but in addition, we enforce that the variables $x$ and $y$ are integers.
> > >
> > > > Another question: if one applies subsampling, it often achieves a worse accuracy. How does it affect the accuracy in your experiments?
> > >
> > > Yes, we agree that if subsampling is used to find a solution for the larger dataset, the accuracy is affected by the sample size. However, our comparisons are over the subsampled part and not the larger dataset. For all of the experiments that compare the objective function, our sample is the first 500 datapoint of the corresponding dataset.
> > > > Finally, my main concern is still about the practical scalability of the algorithm. Even after applying the trick that I suggested, it still needs 5s - 10s to finish a 200-point instance…
> > >
> > > Regarding scalability, we would like to emphasize that none of the scalable methods (for example, Lloyd heuristic) give a theoretical guarantee for this problem. We believe it would be a nice future research direction to devise scalable algorithms with theoretical guarantees for socially fair clustering. The goal of our paper was to devise the first constant-factor approximation algorithm, devise a bicriteria algorithm that outputs only a constant number of extra centers (for fixed $m$), and demonstrate the effectiveness of these algorithms in practice compared to previously best theoretical results.
> > >
> > > We finally would like to emphasize that most of the running time for our method is spent on solving the first original linear program (LP1). Therefore any algorithm based on LP solving (even solving one LP) will have a running time about ours. Moreover, we believe our experiments are slow because we are using solvers that are free to use for academics on a personal laptop. We believe a commercial LP solver on a more sophisticated machine would significantly decrease the running time of our approach. Therefore we believe comparing the running time of our algorithm to other provided baselines is a better way of evaluating it as opposed to only looking at the running time of our algorithm in absolute terms.

---

### Official Review · Reviewer_hqAh · 2022-10-24

**Confidence:** 4
**Correctness:** 4
**Technical Novelty And Significance:** 2
**Empirical Novelty And Significance:** 2
**Recommendation:** 6

**Clarity, Quality, Novelty And Reproducibility:**

I think the paper is quite well written. In general it does a good job covering the related work. The novelty is unclear especially in the first key result on the bi-approximation. I am not quite familiar with the Krishnaswamy et al. paper but from what I read the current result seems incremental w.r.t the former.

**Strength And Weaknesses:**

The bi-approximation result is based on an iterative rounding technique developed by Krishnaswamy et al. Over the iterations, the LP is modified by changing relevant constraints and finally they obtain an LP that is shown to be an intersection of a matroid polytope and m affine spaces. Thanks to a classic result by Grandoni, this implies that the optimal solution for this LP has a support size k +m. Simply rounding these positive fractional variables yields a solution of size k+m centers.

This method seems to be essentially a combination of existing techniques/results, and therefore I would really appreciate the distinctions/comparisons between the current approach and that of Krishnaswamy et al. If it’s just applying the techniques known techniques to a slight generalization – the result would be rather incremental in my opinion.

The sparsification appears to need some careful enumeration but I am not able to judge the importance of the result.


**Summary Of The Paper:**

The problem considered is a “socially fair” variant of (l_p,k)-clustering algorithm. Specifically, the goal is to compute k points, given N points grouped into m classes such that the max weighted sum (for a given weight vector) of L_p metric distances of each group  is minimized over all groups. The central result is a bi-approximation algorithm for this problem where a const^p approximation guarantee is achieved at the expense of m additional centers.

The key prior results for this problem include a:

1.	Const^p*log m approximation guarantee (Makarichev and Vakilian), with a matching lower bound result (under a complexity assumption)
2.	Biapproximation algorithm with 2^O(p)/epsilon approximation guarantee but using k/(1-epsilon) centers.

The current work improves the second result in the case where m << k.

Further, using sparsification techniques developed by Li and Svenson, they show that the additional centers can be removed at the expense of a quasi-poly time.


**Summary Of The Review:**

Overall, I think while the bi-approximation guarantee result is interesting, the techniques seem to be incremental. On the other hand, the sparsification result is perhaps of some mathematically more interest, but the result may be somewhat pedantic and unimportant.

---

> ### Author Response · Authors · 2022-11-19
> **Response to Reviewer hqAh**
>
> > This method seems to be essentially a combination of existing techniques/results, and therefore I would really appreciate the distinctions/comparisons between the current approach and that of Krishnaswamy et al. If it’s just applying the techniques known techniques to a slight generalization – the result would be rather incremental in my opinion.
>
> We would like to emphasize that the problem considered by Krishnaswamy-Li-Sandeep is completely different than our problem of socially fair clustering. They considered the clustering problem in the presence of outliers. Therefore the problem considered by them has an additional linear constraint lower bound on the summation of $x_{ij}$ variables (i.e., the connection of clients to centers) while our linear constraints are on the cost of different groups. Although the geometry of problems at the end of iterative rounding is similar (intersection of a matroid polytope with halfspaces), the problems are different.
>
> > On the other hand, the sparsification result is perhaps of some mathematically more interest, but the result may be somewhat pedantic and unimportant.
>
> Please note that our algorithm gives the first constant factor approximation for the socially fair clustering problem improving over the previously best approximation factor of $\log(m)/\log\log(m)$.

---

### Official Review · Reviewer_3GL3 · 2022-10-25

**Confidence:** 3
**Clarity, Quality, Novelty And Reproducibility:** The paper is well-written and the nov…
**Correctness:** 4
**Technical Novelty And Significance:** 3
**Empirical Novelty And Significance:** 2
**Recommendation:** 6

**Strength And Weaknesses:**

The main result of this paper is interesting and although some ideas are not novel, they are used in a nice way. It is not clear to me the improvement that we are getting with respect to the previous work. With respect to the number of centers opened, the previous work opens a factor $(1+\epsilon)$ more but this work opens an additive $m$ more centers, this can be more or less depending on the setting, but assuming that $m$ is significantly lower that $k$ makes sense to me. Which means that the result can be considered stronger with respect to the number of centers it opens. From the approximation point of view, it is not clear to me which work is better. What is the hidden factor in the approximation guarantee of Abbasi et al?

The ideas in the second and third results are also interesting and the sparsification is nice. I do not think these results are as strong as the main result but definitely a good additional result.

The main weakness of this paper is the experiments, the main concern being the size of the datasets. The size of the considered datasets are rather small and in some cases the algorithm is slower than the baseline. Is it possible to run the bi-critria algorithm of larger datasets, say 100K points or more? The cost improvement over the baselines are significant and interesting. I think it helps to add the exact number of the centers opened by each algorithm in the main body of the paper (it can be added to the same plot as cost as well).


**Summary Of The Paper:**

This paper focuses on the socially fair k-clustering problem. The problem is well-motivated and considers a fair version of the three important and interesting clustering problems, i.e., k-median, k-means and k-center. The authors present four results:

1- A bi-criteria $(9.9)^p$-approximation algorithm that opens (k+m) centers.
2- A $(9.9)^p$-approximation algorithm with running time $O(n^{2^{O(p)} m^2})$.
3- A $(30)^p$-approximation algorithm with running time $k^m poly(n)$.
4- Empirical comparison of their algorithm with the baselines.


**Summary Of The Review:**

The algorithm and theoretical results in good, but there is a concern regarding the performance of the algorithm in larger datasets.

---

> ### Author Response · Authors · 2022-11-19
> **Response to Reviewer 3GL3**
>
> > The main result of this paper is interesting and although some ideas are not novel, they are used in a nice way. It is not clear to me the improvement that we are getting with respect to the previous work. With respect to the number of centers opened, the previous work opens a factor $(1+\epsilon)$ more but this work opens an additive $m$ more centers, this can be more or less depending on the setting, but assuming that $m$ is significantly lower that $k$ makes sense to me. Which means that the result can be considered stronger with respect to the number of centers it opens.
>
> Thanks for pointing out the strength of our approach for the setting where k is larger than $m$. In general, one way to compare the two methods is to set $\epsilon=m/k$. In this case, both methods have a hardbound of k+m centers. Then the previous work gives an approximation factor of $2k/m$ for the fair k-median problem, while our approximation factor is constant.
>
> > The ideas in the second and third results are also interesting and the sparsification is nice. I do not think these results are as strong as the main result but definitely a good additional result.
>
> Although the sparsification techniques for improving the approximation factor constant might not be practical, we believe they are valuable from the theoretical point of view.
>
> > The main weakness of this paper is the experiments, the main concern being the size of the datasets. The size of the considered datasets are rather small, and in some cases, the algorithm is slower than the baseline. Is it possible to run the bi-critria algorithm of larger datasets, say 100K points or more?
>
> We would like to emphasize that the running times of our bicriteria algorithm are only slightly more than the previous bicriteria algorithm by Abbasi-Bhaskara-Venkatasubramanian while we achieve significantly better objective values. The size of datasets is chosen, similar to the previous literature on fair clustering. For example, see [15]. The approaches based on solving linear programs are essentially slower than heuristics such as Lloyd and fair-Lloyd [18] and require more memory. However, we would like to emphasize that the mentioned heuristics do not give a theoretical guarantee on the approximation factor. Unfortunately, we do not think that running the algorithms on 100K points is viable with academic resources and LP solvers at our disposal. However, according to the suggestion of Reviewer 8wgV, we replaced the combinatorial exhaustive search for picking the best $k$-subset out of the selected $k+m$ centers with a mixed-integer linear program that is solved exactly. This significantly improved the running time of our algorithm, which returns exactly $k$ centers for the dataset with $10$ groups. With this new approach, our algorithm is significantly faster than the algorithm of Makarychev-Vakilian on all datasets. Please see the new running times on the rebuttal revision that we have submitted.
>
> > The cost improvement over the baselines are significant and interesting. I think it helps to add the exact number of the centers opened by each algorithm in the main body of the paper (it can be added to the same plot as cost as well).
>
> Thanks for the suggestion! We will consider adding the number of centers to the cost plots if it can be done in a clean way.

---

> > ### Comment · Reviewer_3GL3 · 2022-11-24
> > **Response**
> >
> > 1- The authors mentioned that "Although the sparsification techniques for improving the approximation factor constant might not be practical, we believe they are valuable from the theoretical point of view." Why is that? What is the novelty of this approach? I find it very similar to standard techniques.
> >
> > 2- As pointed out by some of the reviewers, these experiments show mostly that the algorithm cannot be used in practice. Authors mentioned  that " This significantly improved the running time of our algorithm, which returns exactly  centers for the dataset with  groups", is this tried on larger datasets? Currently I do not see how these algorithms can be used in practice. They only work for very small instances.
> >
> > 3- I think this is not a good reason "The size of datasets is chosen, similar to the previous literature on fair clustering". We have results in clustering context that the size of instance in the new works are 2-3 order of magnitude higher than previous works just to show that their approach can be used in practice.
> >
> > 4- I understand that the focus of this paper is mostly on theoretical results but we need to keep in mind that the theoretical results here are also not that strong. The ideas are nice and novelty is not impressive but good, I do not think this paper is a clear accept based on its theoretical result.
> >
> > Overall my concerns are the same as before and I still think the experiments in this work are weak. I keep my score.

---

> > > ### Author Response · Authors · 2022-11-25
> > > **Novelty and Experiments**
> > >
> > > We would like to summarize our theoretical contributions as follows. We simplify and adapt the iterative rounding framework in a novel setting and prove its approximation guarantee. We combine this with sparsification techniques to give the first constant factor approximation. The adaption of sparsification techniques requires care since the objective values of individual groups are different. This enforces picking sparser instances. Note that the sparsification is highly non-trivial given our objective of the maximum over the costs of groups. For example, if there is only one group (classic $k$-median), the statement of Lemma 4.1 is trivial, but in our case, it is not, and it requires careful enumeration. While we definitely build on known techniques, the extension, combination, and the results obtained are novel.
> > >
> > > Regarding the size of datasets, while our methods (and methods in the literature) are not yet at the speeds of standard (non-fair) approaches for $k$-means clustering, we believe that our theoretical and empirical results together demonstrate the feasibility of these methods in practice for the important problem of fair clustering. We anticipate that future work will find even faster and more practical methods.

---

### Decision · Program_Chairs · 2023-01-20

**Decision:**

Reject

**Justification For Why Not Higher Score:**

Novelty and scalability issues: this genre of research is important and has been attracting a good number of papers, hence the acceptance threshold for such research has gone up.

**Justification For Why Not Lower Score:**

N/A

**Metareview: Summary, Strengths And Weaknesses:**

This paper considers the increasingly important notion of demographic fairness in a general clustering context, where we need to open k “facilities” (cluster centers) in order to minimize the ell_p norm of the distances to the cluster center for each of the points in the data-set (or the p’th power of the ell_p norm). The added fairness ingredient is that we are given some demographic groups, and we aim to minimize the maximum cost (total distance traveled, or its ell_p norm to the power of p) over all of the given groups. I believe this sort of algorithmic research is important in fair implementations of ML.

The main concerns raised were novelty and run time: since more papers are being written in this general area in AI/ML, the acceptance threshold for papers in this sub-field has gone up significantly. As the authors point out, the latter is due to the time taken to solve a large linear program (LP). While this is understandable (and while it is also true that faster approaches do not have good theoretical guarantees like in this work), the authors could perhaps work on combinatorial approximation schemes for such LPs---for instance, perhaps using multiplicative weights update, as has been done recently for many classes of LPs---and also see if the LP itself can get sparsified rigorously.


**Summary Of Ac-Reviewer Meeting:**

N/A